# Generative Adversarial Post-Training Mitigates Reward Hacking in Live Human-AI Music Interaction

**Yusong Wu[1], Stephen Brade[3], Aleksandra Teng Ma[4], Tia-Jane Fowler[5], Enning Yang[6], Berker Banar[7], Aaron Courville[1,2], Natasha Jaques[5]\*, Cheng-Zhi Anna Huang[3]\***

[1]Mila, Quebec Artificial Intelligence Institute, Université de Montréal
[2]Canada CIFAR AI Chair
[3]Massachusetts Institute of Technology
[4]Georgia Institute of Technology
[5]University of Washington
[6]McGill University
[7]Independent Researcher
wu.yusong@mila.quebec, nj@cs.washington.edu, huangcza@mit.edu

## Abstract

Most applications of generative AI involve a sequential interaction in which a person inputs a prompt and waits for a response, and where reaction time and adaptivity are not important factors. In contrast, live jamming is a collaborative interaction that requires real-time coordination and adaptation without access to the other player's future moves, while preserving diversity to sustain a creative flow. Reinforcement learning post-training enables effective adaptation through on-policy interaction, yet it often reduces output diversity by exploiting coherence-based rewards. This collapse, known as "reward hacking", affects many RL post-training pipelines, but is especially harmful in live jamming, where musical creativity relies on dynamic variation and mutual responsiveness. In this paper, we propose a novel adversarial training method on policy-generated trajectories to mitigate reward hacking in RL post-training for melody-to-chord accompaniment. A co-evolving discriminator separates policy trajectories from the data distribution, while the policy maximizes the discriminator output in addition to coherence rewards to prevent collapse to trivial outputs. We evaluate accompaniment quality and output diversity in simulation with both fixed test melodies and learned melody agents, and we conduct a user study with the model deployed in a real-time interactive system with expert musicians. Quantitative evaluation and user feedback demonstrate improved output diversity, harmonic coherence, adaptation speed and user agency. Our results demonstrate a simple yet effective method to mitigate reward hacking in RL post-training of generative sequence models.[1]

## 1 Introduction

The combination of large-scale transformer-based models and reinforcement learning (RL) post-training has revolutionized AI, with over 1 billion people now using large language models (LLMs) trained with these techniques (OpenAI, 2025; Perez, 2025). However, most applications of generative AI still involve a slow back-and-forth interaction, where the user inputs a request, and then waits—sometimes several minutes—for a response. Further, RL post-training techniques are limited by their vulnerability to "reward hacking", where the policy exploits the reward to produce either unintelligible outputs or trivial, low-diversity content that scores well yet fails to engage users (Skalse et al., 2022; Lewis et al., 2017).

---

\*Equal contribution as senior authors.
[1]Code: https://github.com/lukewys/realchords-pytorch
Audio examples: https://realchords-GAPT.github.io

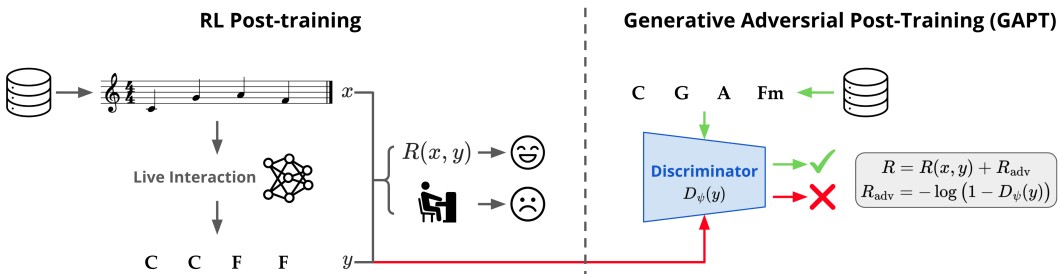

Figure 1: **Left:** RL post-training enables real-time adaptation for melody-to-chord accompaniment but is vulnerable to reward hacking: the policy exploits the coherence reward $R(x,y)$ by repeating simple, high scoring chords, which reduces diversity and breaks creative flow. **Right:** We propose an adversarial reward signal to prevent reward hacking. A discriminator $D_\psi(y)$ trained to distinguish policy rollouts from data, with its realism estimation added to the reward. This regularizes the policy toward natural accompaniment while preserving input coherence, preventing diversity collapse.

Live music jamming is a collaborative interaction requiring real-time coordination and adaptation without seeing the partner's future moves, and demanding creative expression supported by diversity. Imagine stepping onto a stage to jam with a musician you have never met. Within the first few beats, you establish alignment, plan your next move without knowing the future moves of your partner, anticipating the partner to maintain coherent harmony, and recover quickly from errors (Keller, 2008), all while keeping progressions varied to support creative flow (Wrigley & Emmerson, 2013) and emotional connection (Trost et al., 2024).

Training generative music models that can jam live with humans is rare in existing works and demanding from both engineering and algorithmic perspectives, as such systems must run at low latency, recover from errors, and adapt on the fly without access to future input. Supervised maximum likelihood training is straightforward, yet such models often fail at deployment because curated corpora are well composed and rarely contain mistakes, corrective maneuvers, or co-adaptive behavior (Wu et al., 2024; Jiang et al., 2020). Reinforcement Learning (RL) post-training offers a promising alternative by simulating interactive sessions and optimizing rewards on accompaniment coherence (Wu et al., 2024; Jiang et al., 2020). However, optimizing a learned reward can induce behavior similar to RL post-training of dialogue models, where the policy maximizes a reward parameterized by another learned model (Skalse et al., 2022). This leaves it vulnerable to reward hacking, where it discovers outputs that adversarially trick the reward model into assigning spuriously high scores for bad inputs. In dialogue, this can sometimes look like manipulating the reward model into thinking it has satisfied a user's preferences when it has not. In music, this appears as non-varying repetitive accompaniment that is highly harmonic, but simple (Fig. 2); it diminishes user experience by reducing perceived control and agency in creative jamming (Fig 3). This type of "reward hacking" behavior is an inherent limitation of RL post-training, and can be frequently observed in dialogue applications as well (Wan et al., 2025).

In this paper we propose **Generative Adversarial Post-Training** (GAPT, Fig. 1), an adversarial augmentation of RL post-training for real-time live music jamming inspired by Generative Adversarial Networks (Goodfellow et al., 2014; Ho & Ermon, 2016). Specifically, we target melody-to-chord accompaniment, where a policy generates chords online in response to a live melody stream without access to the partner's future moves. Alongside the coherence-based task reward, we train a discriminator to distinguish policy trajectories from data. As in GANs, the discriminator is updated online as the policy improves to get progressively better at making fine distinctions between the generated samples and the real data (Goodfellow et al., 2014). Unlike GANs, gradients cannot be backpropagated through sequence sampling, so we use RL to optimize the output signal of the discriminator, in the spirit of Generative Adversarial Imitation Learning (Ho & Ermon, 2016).

However, adversarial training is effective only if the policy can effectively increase the discriminator's "realism" estimation. This is difficult because discriminator updates make the induced reward nonstationary, and an overpowered discriminator impedes the policy from optimizing the reward signal (Arjovsky et al., 2017). We therefore introduce a two-phase training schedule that yields a stable, easy-to-optimize reward. Phase one warms up the discriminator with fixed-interval updates, while

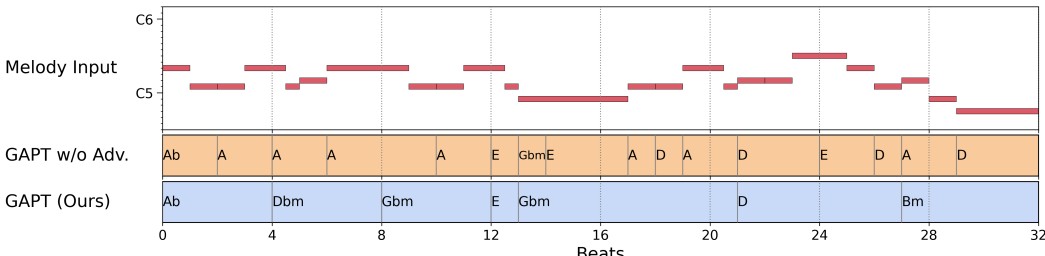

Figure 2: Under the same melody input stream (first row) in a live accompaniment setting, the model trained without adversarial reward (second row) produces harmonically coherent yet unnatural progressions with repetitive, trivial, and low-coverage chord choices that hinder human-AI interaction. In contrast, `GAPT` (third row) produces coherent, natural, and diverse live chord accompaniment by jointly training the policy with a discriminator that supplies an adversarial reward.

phase two applies adaptive updates: we update the discriminator only when the moving average of discriminator reward over recent policy steps exceeds a threshold, and keep it frozen otherwise.

Together, the discriminator and the coherence reward provide complementary constraints that mitigate reward hacking: repetitive, low-variation progressions that exploit the coherence reward can be easily identified, resulting in low realism reward, whereas outputs that chase realism but ignore the live melody receive poor coherence reward. The two-phase adaptive discriminator update balances learning speeds, reduces oscillation, and supports steady growth in reward. In this way, `GAPT` plays a similar role to the KL constraint commonly used in RL post-training (Jaques et al., 2017). However, we find that the KL constraint is insufficient to mitigate reward hacking in our setting. Adversarial training is necessary to preserve realism while learning to adapt and optimize for accompaniment coherence across diverse inputs, unlocking the benefits of RL post-training for real-time applications demanding robust, adaptive responses to many users.

To evaluate the effectiveness of `GAPT`, we simulate live interaction with two sources of input: fixed test set melodies and a learned counterpart agent. We then deploy the trained policy in a real-time interactive system and conduct a user study with 12 expert musicians. We evaluate accompaniment quality and diversity on the generated music across all three settings, and in the user study we additionally collect quantitative participant ratings. Across the extensive evaluations, our method improves adaptability, preserves diversity, and increases perceived user agency compared with pure RL post-training baselines.

The contributions and findings of our research are summarized as follows:

- We propose **Generative Adversarial Post-Training** (GAPT), a method that mitigates reward hacking when fine-tuning Transformer-based generative sequence models. It uses a two phase joint optimization of the policy and a discriminator, where the discriminator is first warmed up, then updated adaptively only when the policy meaningfully increases the realism reward.
- We apply `GAPT` to the challenging setting of real-time musical accompaniment, which demands both harmonic coherence and progression diversity without access to the partner's future moves.
- We evaluate the model in simulation and deploy it in a real-time system that interacts live with expert musicians. The adversarially augmented model improves diversity, harmonic coherence, adaptation speed, and perceived agency over baselines.
- We release training datasets, model checkpoints, and code for an RL training infrastructure for Transformer-based music agents and for the real-time interactive system.

## 2 RELATED WORKS

**Reward Hacking in RL Post-training**  RL post-training is widely used to align pretrained generative sequence models with human preferences (Ouyang et al., 2022). However, research has found that maximizing a learned reward can induce models to exploit weaknesses in the reward signal (Gao et al., 2023), resulting in trivial, repetitive outputs that obtain high scores but drift from natural syntax and semantics, or do not correspond to desired behavior (Skalse et al., 2022; Lewis et al., 2017). This phenomenon is often referred to as "reward hacking" (Skalse et al., 2022), also called

"alignment tax" (Askell et al., 2021) or "language drift" (Lee et al., 2019). A common mitigation constrains deviation from the pretrained model by adding a Kullback-Leibler (KL) penalty to the training objective (Jaques et al., 2017; Ouyang et al., 2022; Lee et al., 2023). Recent work finds that the KL constraint can be insufficient and proposes additional approaches such as elastic reset (Noukhovitch et al., 2023), reward shaping (Fu et al., 2025; Yan et al., 2024), personalized rewards (Wan et al., 2025). Orthogonal to these policy-side mitigation, Bukharin et al. (2025) introduce adversarial training in the reward model to mitigate reward hacking. In this work, we propose a simple yet effective mitigation by training a discriminator that provides adversarial rewards together with policy regularizing. The discriminator acts as a regularizer, effectively constraining the policy from collapsing to trivial outputs.

**Real-time Music Accompaniment Systems** Early real-time accompaniment systems relied on non-neural methods, divided into score following and rule-based or corpus-based generation. Score-following approaches align a performance to a fixed score and play pre-defined accompaniment accordingly (Dannenberg, 1984; Raphael, 2010; Cont, 2008). Generative and co-creative systems synthesize accompaniment using rules or by recombining material learned from a corpus predefined or constructed on the fly (Lewis, 2003; Assayag et al., 2006; Nika & Chemillier, 2012; Nika et al., 2017). Recent deep learning systems model musical context and interaction directly. Early systems primarily used supervised maximum likelihood, for example LSTM-based BachDuet (Benetatos et al., 2020) and the Transformer-CRF pipeline in SongDriver (Wang et al., 2022). Recently, systems adopt RL post-training to improve adaptability by optimizing on interactive trajectories. RL-Duet (Jiang et al., 2020) pioneered this approach, employing RL fine-tuning to adaptively generate music. ReaLchords (Wu et al., 2024) introduced transformer-based models fine-tuned via RL with knowledge distillation and self-supervised coherence reward, and ReaLJam (Scarlatos et al., 2025) constructs a real-time human-AI jamming interface based on ReaLchords model. Building upon ReaLchords, we diagnose diversity collapse under coherence-only RL as reward hacking and propose a novel Generative Adversarial Post-Training that mitigates reward hacking while preserving coordination. Moreover, we extend evaluation beyond model-data simulation to include model-model interaction and a human study with musicians using a real-time interactive system.

**Generative Adversarial Learning** Generative adversarial learning trains a generator and a discriminator in a two-player game, where the generator aims to produce samples that the discriminator cannot distinguish from data (Goodfellow et al., 2014). For more than half a decade from 2014-2020, this paradigm was successfully applied to improve images (Karras et al., 2020), audio (Kumar et al., 2019), and text (Yu et al., 2017). In RL, similar generative adversarial objectives were widely used for off-policy imitation learning, where a discriminator-induced reward aligns the policy's occupancy measure with expert demonstrations, as in Generative Adversarial Imitation Learning (GAIL) (Ho & Ermon, 2016) and Adversarial Inverse Reinforcement Learning (AIRL) (Fu et al., 2017). However, for the most part these approaches have fallen out of favor in the modern post-LLM era, with recent works use adversarial training merely for sequence generation on specific task (Zhang et al., 2024; Wang et al., 2024; Yu et al., 2023). Peng et al. (2021) trains a discriminator along with a robotic control policy to increase the naturalness of the motion, but their application is limited to robotics with simple policy model. Our work shows that they still hold value for mitigating reward hacking, and demonstrates their effectiveness for a challenging real world live interaction task.

## 3 METHODS

### 3.1 BACKGROUND

We study collaborative music co-creation where two agents (either a trained model or a human player) act concurrently to produce a joint sequence $(x_1, y_1), \ldots, (x_T, y_T)$. At each discrete step $t$, both agents observe the shared history $x_{<t}, y_{<t}$ and simultaneously emit the next melody token $x_t$ and chord token $y_t$. We define the simultaneous generation process as conditionally independent given the shared history:

$$\Pr(x_t, y_t \mid x_{<t}, y_{<t}) = \Pr(x_t \mid x_{<t}, y_{<t}) \Pr(y_t \mid x_{<t}, y_{<t}). \tag{1}$$

In the general setting, the melody $x$ and chords $y$ co-involve via their shared history $(x_{<t}, y_{<t})$, corresponding to musicians adapting to each other as they play. As a first step, we focus on the

accompaniment setting where the model is trained with melody $p(x_t \mid x_{<t})$ fixed. The accompaniment setting assumes the melody is taking the lead and the chord is following the melody. We also assume a "cold-start" coordination where the two agents do not have prior knowledge of each other and do not have shared context at the beginning of the accompaniment. We train a chord-generation policy $\pi_\theta$ for real-time accompaniment with the following *online* dependency:

$$\pi_\theta(y \mid x) = \prod_{t=1}^{T} \pi_\theta\big(y_t \mid x_{<t}, y_{<t}\big). \tag{2}$$

The factorization in Eq. 2 enforces the online constraint, since $\pi_\theta\big(y_t \mid x_{<t}, y_{<t}\big)$ does not depend on $x_t$ or any future tokens, which enables the model to be deployed live, to jam in real-time with a person. We first pretrain $\pi_\theta$ by maximum-likelihood estimation (MLE) on paired melody and chord sequences from a dataset $\mathcal{D}$ of $(x, y)$ pairs:

$$\max_\theta \ \mathbb{E}_{(x,y)\sim\mathcal{D}} \left[ \sum_{t=1}^{T} \log \pi_\theta\big(y_t \mid x_{<t}, y_{<t}\big) \right]. \tag{3}$$

Purely supervised online models trained on $\mathcal{D}$ often fail at deployment due to exposure bias (Wu et al., 2024). Curated datasets of composed scores rarely include mistakes, corrective maneuvers, or co-adaptive behavior, so the model does not practice recovery or adaptation. During inference, the policy conditions on its own past outputs, which leads to error accumulation and out-of-distribution states if it never practices recovery after a misprediction or adapts to changes in the input distribution.

## 3.2 REINFORCEMENT LEARNING POST-TRAINING

RL post-training equips the melody-to-chord accompaniment policy with two online skills that MLE alone lacks: anticipating upcoming inputs and adapting to distributional changes during interaction. In particular, by sampling from its own policy to produce rollouts and updating on their successes and failures, RL reduces the mismatch between training and deployment and is less vulnerable to encountering out-of-distribution states when generating from its own outputs. To implement RL post-training for music generation, we sample a batch of melodies $x$ from the dataset and roll out the policy online according to Eq. 2, producing a chord trajectory $y$. Unlike pairs from the curated dataset, on-policy trajectories $(x, y)$ naturally include adaptation, recovery, and mis-anticipation events that arise during learning. Afterwards, we compute a scalar episode reward $R(x, y)$ using an ensemble that combines (i) self-supervised coherence rewards (§3.4), (ii) rule-based penalties (§3.4), and (iii) an adversarial reward signal that scores how data like the generated chord trajectory is (§3.3). The policy is then updated by optimizing a KL-regularized objective:

$$\max_\theta \ \mathbb{E}_{x\sim\mathcal{D},\, y\sim\pi_\theta(\cdot|x)} \Big[ R(x, y) - \beta\, D_{KL}(\pi_\theta(\cdot \mid x) \parallel \phi_\omega(\cdot \mid x)) + \gamma \sum_{t=1}^{T} \mathcal{H}\big(\pi_\theta(\cdot \mid x_{<t}, y_{<t})\big) \Big], \tag{4}$$

where $\sum_{t=1}^{T} \mathcal{H}\big(\pi_\theta(\cdot \mid x_{<t}, y_{<t})\big)$ is an entropy regularization loss to promote output diversity; $\beta$ and $\gamma$ are coefficients balancing the KL objective and the entropy loss. We use Proximal Policy Optimization (PPO) (Schulman et al., 2017) for the reward maximization objective. We initialize both the policy and the value model in PPO from the MLE pretrained checkpoint. Following prior work (Wu et al., 2024), the KL anchor $\phi_\omega$ is a trained offline model that conditions on the full input:

$$\phi_\omega(y \mid x) = \prod_{t=1}^{T} \phi_\omega\big(y_t \mid x, y_{<t}\big). \tag{5}$$

One common practice in RL post-training of LLMs is to use the initialization of policy as KL anchor to prevent reward hacking. However, previous work (Wu et al., 2024) shows that RL finetuning with MLE KL anchor fails to train a good model in the online accompaniment setting.

## 3.3 GENERATIVE ADVERSARIAL POST-TRAINING

To counter reward hacking, we introduce a discriminator $D_\psi(\cdot)$ implemented as a Transformer encoder that maps a policy-generated trajectory $y$ to a realism estimation $D_\psi(y)$. During on-policy RL

post-training, $D_\psi$ co-evolves with the policy via a binary classification: sequences from the dataset are labeled positive, and sequences $\hat{y}$ generated by the current policy while interacting with the input are labeled negative. In the interactive setting, we train $D_\psi$ only on the outputs of the model not the full interaction trajectory so that it captures an input-agnostic prior that transfers to unseen inputs.

We incorporate the discriminator signal into the policy objective by rewarding sequences that it deems data-like. Following (Ho & Ermon, 2016), we define an adversarial reward $R_{\mathrm{adv}} = -\log(1 - D_\psi(y))$, where $D_\psi(y) \in [0, 1]$ is the discriminator's estimate that the policy-generated chord trajectory $y$ comes from the data distribution. The task reward and the adversarial reward create complementary pressures: sequences that achieve high task reward by exploiting unrealistic shortcuts result in low realism estimation and are penalized through $R_{\mathrm{adv}}$, while sequences that appear data-like yet fail to optimize the task objective receive low task reward. This combination steers the policy toward diverse, well-formed and realistic outputs that follows data distribution by adaptively penalizing reward-hacking behavior during training.

A practical challenge is that updating $D_\psi$ jointly with the policy introduces constraints that hinder learning stability. We know from the rich history of work on GANs that if $D_\psi$ advances too quickly, the policy receives vanishing or uninformative gradients, which impedes reward maximization (Arjovsky et al., 2017). Because $D_\psi$ is updated throughout training, the reward signal is also nonstationary, which destabilizes optimization. We therefore adopt a two-phase update schedule with adaptive discriminator update. Phase 1 performs a short warmup with a fixed update ratio to roughly match learning speeds: one $D_\psi$ update after every five PPO policy updates for the first 200 steps. Phase 2 switches to adaptive, confidence-gated updates that address both constraints. Let $\bar{R}_{\mathrm{adv}}$ be the moving average of the adversarial reward over the most recent three PPO updates. We enable a discriminator step when $\bar{R}_{\mathrm{adv}} > \tau$ (we use $\tau = 1.0$), otherwise we keep $D_\psi$ frozen. Gating holds $D_\psi$ static when its signal would be unstable or overpowering, then advances $D_\psi$ only once the policy has caught up enough for the reward signal to be informative. To reduce overfitting in the discriminator, we apply label smoothing with $\alpha = 0.1$ to the binary cross-entropy targets (Szegedy et al., 2016). See Alg. 1 for a pseudocode of the training process.

## 3.4 REWARD MODELS AND REGULARIZATION

**Self-supervised coherence rewards**  Following and extending ReaLchords (Wu et al., 2024), we construct $R(x, y)$ from an ensemble of self-supervised rewards and compute a single per-episode score for each rollout. We do not use multi-scale rewards as in ReaLchords. All reward models are trained and evaluated at the full sequence length, equal to the maximum context of the accompaniment model. A contrastive model encodes melody and chord into embeddings and is trained with an InfoNCE objective (Oord et al., 2018; Radford et al., 2021) to align true pairs within a batch. At evaluation time, the cosine similarity between the melody and chord embeddings provides a global harmonic-alignment signal. A discriminative model takes the full pair $(x, y)$ and outputs the probability that the pair is real rather than a randomly re-paired negative, which provides a complementary temporal-coherence signal. To mitigate bias introduced by pitch-shift augmentation, we include rhythm-only variants of both models that strip pitch and retain onset, hold, and silence tokens. For each reward type and input variant, we train two seeds and ensemble their normalized scores, then average across all components to obtain the per-episode reward $R(x, y)$.

**Rule-based penalties**  We include four penalties during RL to regularize training: (i) an invalid output penalty for format violations, (ii) a silence penalty applied when more than $4\%$ of frames are silent while the melody is active, with a grace period covering the first 8 frames of each sequence, (iii) an early stop penalty when an end of sequence token is emitted before the melody ends, and (iv) a repetition penalty when the same chord repeats for more than four consecutive times.

## 4 EXPERIMENTS

**Overview**  We evaluate whether the GAPT improves adaptation to live input while preserving diversity. We consider three settings that progressively increase interactivity: (i) fixed melody simulation, where the accompaniment policy responds online to held-out melodies; (ii) model–model interaction, where a learned melody jamming agent co-adapts with the chord policy, which better approximates playing with a human partner who adapts online to the accompaniment and probes

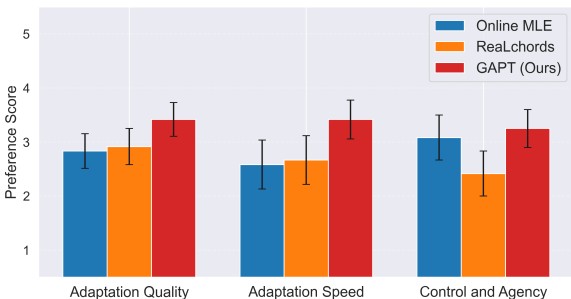

Figure 3: Participant ratings for real-time jamming with each model. Error bars show standard error. `GAPT` has the highest mean on all three evaluation questions and significantly improves adaptation speed and perceived control and agency over ReaLchords ($p < 0.05$). The improved user experience benefits from higher diversity under generative adversarial post-training.

mutual adaptation; and (iii) a real-time user study with expert musicians jamming with the model deployed in an interactive system. We report adaptation quality, output diversity, and user ratings. We report model architecture and training details in §C.

### 4.1 DATASET AND DATA REPRESENTATION

We train our models on three datasets: Hooktheory (Donahue et al., 2022), Nottingham (Allwright et al., 2003), and POP909 (Wang* et al., 2020). All datasets comprise monophonic melodies and chords from pop or folk songs. We apply data augmentation by randomly transposing each piece by $k$ semitones with $k \in \{-6, \ldots, 6\}$. For evaluation, we additionally evaluate the policy on the Wikifonia dataset (Simonetta et al., 2018), which is excluded from training. The details of dataset are included in §B.2

We follow the frame-based representation used in ReaLchords Wu et al. (2024). For the melody $\{x_1, \ldots, x_T\}$ and the chord sequence $\{y_1, \ldots, y_T\}$, each time step $t$ corresponds to a time quantized to sixteenth note frame and carries a pair $(x_t, y_t)$ of discrete tokens. Melody tokens use the vocabulary of ON_{p} and HOLD_{p}, where $p$ is the MIDI pitch active at frame $t$. We emit ON_{p} on the onset frame of a note with pitch $p$, and HOLD_{p} on subsequent frames until that note ends. Chord tokens use the similar vocabulary of ON_{c} and HOLD_{c}, where $c$ is a chord symbol. We set a maximum sequence length $T \leq 256$ for both $x$ and $y$. To respect the online dependency in Eq. 2, the policy is trained on an interleaved stream $\{y_1, x_1, y_2, x_2, \ldots, y_T, x_T\}$.

### 4.2 SYSTEMS COMPARED

**Online MLE.** The accompaniment policy trained only with MLE via Eq. 3 (supervised learning). **ReaLchords.** A reproduction of Wu et al. (2024) trained with the ensemble of coherence rewards and penalties (§3.4), no entropy term ($\gamma = 0$), and trained only on the Hooktheory dataset. **GAPT.** Our method: on-policy PPO with the ensemble of coherence rewards and penalties (§3.4) combined with adversarial reward (§3.3) and the entropy term. **GAPT w/o Adv. Training.** An ablation that removes the adversarial reward and keeps all other components identical. In the user study we compare Online MLE, ReaLchords, and `GAPT`.

### 4.3 EVALUATION SETTINGS

**Fixed melody simulation** We stream each held-out melody $x$ and roll out the accompaniment policy online according to Eq. 2 to obtain $y$. This setting isolates online adaptation to real melodies without a co-adapting partner. We evaluate both on combined test set of three datasets used to train the model, as well as on an out-of-distribution dataset Wikifonia that is never seen during training.

**Model-model interaction** To study co-adaptation, we train a melody jamming agent that generates melody given shared history: $\pi^x(x \mid y) = \prod_{t=1}^{T} \pi^x(x_t \mid x_{<t}, y_{<t})$. We train this melody jamming agent on all three datasets used for training the chord policy, as well as the unseen Wikifonia dataset. This is because we would like to test the generalization of the chord agent to a partner that has novel musical styles it may not have encountered. Our goal is to create a simulated adaptive agent with different experience than the chord agent to better estimate how the chord agent would perform with real human musicians. We train the melody jamming agent using the same RL objective and reward formulations as the chord policy, as well as the adversarial reward. We also train an

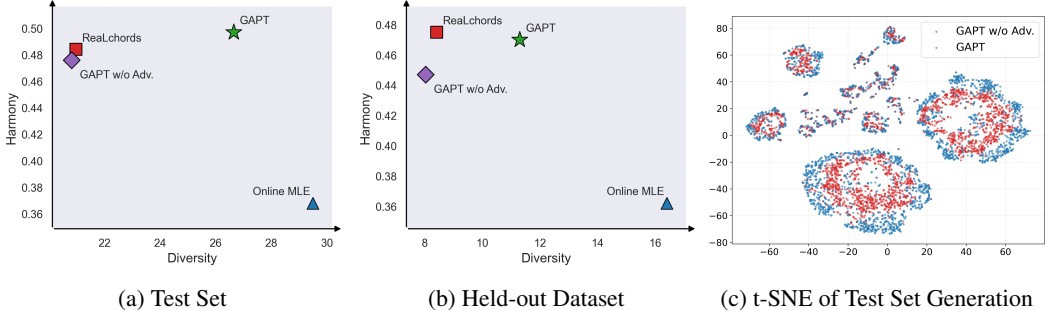

(a) Test Set  (b) Held-out Dataset  (c) t-SNE of Test Set Generation

Figure 4: `GAPT` advances the Pareto frontier for diversity versus harmony. In simulated interaction on the test set (a) and on an out-of-distribution dataset (b), `GAPT` attains higher diversity while preserving strong harmony. By contrast, Online MLE without RL produces diverse outputs but fails at harmonic coherence during interactive generation. ReaLchords and `GAPT` without adversarial training achieves strong harmony at the cost of diversity. The t-SNE visualization of test set generations (c) likewise shows that `GAPT` covers a broader region of the accompaniment space.

offline melody model $\phi^x(x \mid y)$ that conditions on full chord context. During evaluation, the two online policies act simultaneously according to Eq. 1, conditioning on the shared history.

**Real-time Interactive System and User Study**  We deploy a client-server system adapted from ReaLJam (Scarlatos et al., 2025) that generates music in chunks with a fixed lookahead which maintains a buffer of planned outputs to handle network latency (see details in §B.4). We recruit 12 experienced musicians, most with more than 10 years of instrument practice and some with improvisation experience. Each session is in person. After a short demo and a one minute familiarization, participants interact with three anonymized systems in counterbalanced order. For each system, they complete three tasks in increasing adaptability: (1) play a fixed melody, (2) improvise in one key and modulate to a second key midway, and (3) co-improvise while attending to upcoming chords. Each task lasts 1-2 minutes. After each system, participants answer three 5-point Likert-scale items: *Adaptation quality* (the harmony matched my melody), *Adaptation speed* (the model adapted quickly to changes), and *Control and Agency* (I felt that I had control and agency during the session). We report per-question average score across different systems. We also conduct a brief free-form interview at the end.

## 4.4 EVALUATION METRICS

**Adaptation quality: note-in-chord ratio**  We measure adaptation with the note-in-chord ratio, computed as the proportion of frames where the melody pitch class belongs to the concurrent chord. For example, if the melody token at time $t$ has pitch C and the chord token at that time is C major, that frame contributes 1 to the ratio. We evaluate only frames where a melody note is active, then average this indicator across frames and examples to report the overall ratio.

**Diversity: Vendi Score**  We assess chord diversity with the Vendi Score (Friedman & Dieng, 2023; Pasarkar & Dieng, 2024). First, we embed each chord sequence $y^{(i)}$ into a vector $z_i$ using the chord encoder from the contrastive reward model (§3.4). Next, we compute pairwise similarities between embeddings via cosine similarity to form an $N$ by $N$ Gram matrix, where $N$ is the total number of chord sequences. We normalize this matrix and compute the Vendi Score as the Shannon entropy of the eigenvalues of the normalized matrix. The Vendi Score reflects the effective number of distinct patterns in the set, with higher values indicating greater diversity.

We note that neither metric suffices in isolation. Maximizing harmony at the expense of diversity yields repetitive accompaniment, whereas maximizing diversity without harmony yields disorder. Therefore an ideal music generation model should pushes the Pareto frontier of both metrics, maintaining high harmony and diversity even when encountering novel melodies or jamming partners.

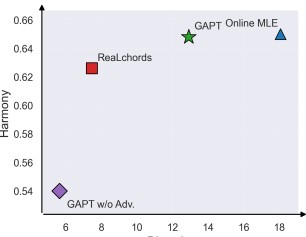 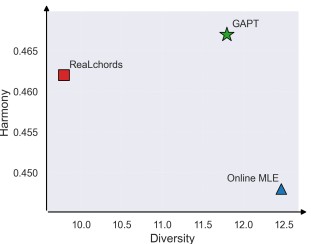

(a) Melody Jamming Agent  (b) User Interaction Trajectory

Figure 6: Harmony and diversity evaluated with a learned melody jamming agent (a) and in live user sessions (b). GAPT preserves harmonic coherence while restoring progression diversity compared to ReaLchords, yielding a better harmony and diversity tradeoff and higher perceived control and adaptation speed.

## 5 RESULTS

**Fixed melody simulation** Figure 4a and Figure 4b report online accompaniment on the test set and a held-out dataset, with detailed numbers in Table 1. The pattern is consistent across both settings. Online MLE yields high diversity but low harmony because it lacks RL post-training. ReaLchords and the ablation without the adversarial reward achieve strong harmony but at the cost of reduced diversity. GAPT excels on both diversity and accompaniment harmony, indicating that the adversarial training specifically recovers diversity without sacrificing coherence. To visualize coverage, we embed test set generations with the reward model's chord encoder and plot t-SNE projections (Figure 4c); GAPT spans a broader region than the non-adversarial ablation, indicating more varied accompaniment. We further experiment with applying perturbation to input melody. GAPT adapts quickly while maintaining harmony, as shown in Figure 7.

**Co-adaptation with Melody Jamming Agent** Figure 4 summarizes co-adaptive interaction, with full results in Table 2. GAPT consistently outperforms ReaLchords and the non-adversarial ablation on both harmony and diversity, supporting that the adversarial reward acts as an explicit diversity regulator during mutual adaptation. A notable exception is Online MLE, which appears strongest when paired with a partner trained to accommodate it; this setting reduces the need for recovery or adaptation and keeps interaction states close to the curated distribution. This effect does not generalize to human partners, as shown later in the user study.

**Real-time user study** Figure 3 shows participant ratings. GAPT achieves the highest mean on all three questions and significantly exceeds ReaLchords on adaptation speed and on perceived control and agency ($p < 0.05$). In complementary analyses on the human interaction trajectories, Figure 5b and Table 2 place GAPT on the empirical Pareto frontier of harmony and diversity, consistent with the simulation results.

**Qualitative feedback** Participants evaluated three anonymized systems corresponding to Online MLE, ReaLchords, and GAPT. Comments highlight the tradeoffs we target:

- On GAPT, *"catches my key and chord changes faster... it would prompt the right chord to resolve the suspension... it definitely adapts faster."* (P10)

- On ReaLchords, *"harmony was fine but it was really dumb... it just keeps giving me the same two chords... it was a little boring."* (P7)

- On Online MLE, *"took some time to adapt... chords not really matching... but sometimes it tried novel progressions."* (P11)

- Direct comparison favored GAPT over ReaLchords: *"I liked the first one better."* (P6, GAPT vs ReaLchords). Another noted GAPT *"was listening... following the kind of thing I expected."* (P3)

Together with Fig. 3, these remarks indicate that the additional diversity from the GAPT yields quicker, more responsive collaboration without collapsing to trivial repetitions.

Across fixed melodies, co-adaptive simulation, and live sessions with musicians, GAPT mitigates reward hacking by preventing diversity collapse while preserving harmony. GAPT attains higher adaptation quality than MLE, higher diversity than coherence-focused RL, and improved perceived speed and agency in human studies.

## 6 CONCLUSION

We studied real-time melody-to-chord accompaniment where RL post-training on coherence rewards tends to collapse output diversity. We introduced a Generative Adversarial Post-training that trains a discriminator to provide data-likeness reward signal during policy optimization, together with a simple two-phase update schedule that stabilizes learning. Across fixed-melody simulation, model-to-model co-adaptation, and a user study with recruited musicians, our method improves harmonic adaptation while restoring diversity to near dataset levels, and yields higher perceived adaptation speed and user agency. These results show that a lightweight adversarial training is an effective and practical mitigation for reward hacking in RL post-training of generative sequence models. Future work includes extending the adversarial training to multi-agent co-adaptive training and integrating personalized preference models.

## ACKNOWLEDGMENT

We thank Kimaya Lecamwasam, Ke Chen, and Heidi Lei for their valuable discussions related to this project. We are also grateful to Kunal Jha, Carrie Yuan, Yamming Wan and Gaurav Sahu for their helpful advice on paper formatting and writing.

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

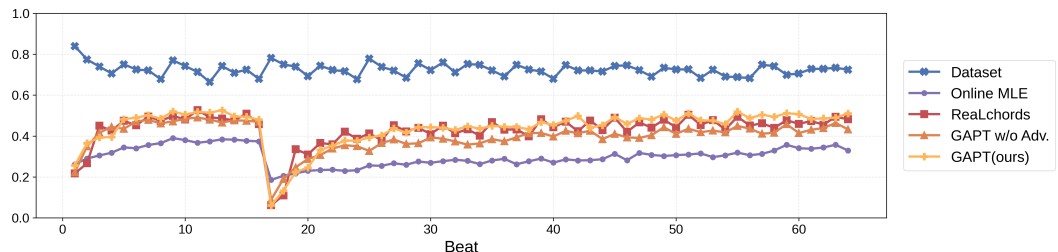

Figure 7: Live accompaniment harmony (note-in-chord ratio) across time. We transpose the test set melody input up 6 semitones at 16-th beat to introduce perturbation. `GAPT` adapts quickly to the perturbation and maintains good harmony afterward.

---

**Algorithm 1** Generative Adversarial Post-Training

---

**Require:** Dataset $\mathcal{D}$, policy $\pi_\theta$, discriminator $D_\psi$, warmup $T_{\text{warm}}$, interval $K_{\text{int}}{=}5$, window $m{=}3$, threshold $\tau = 1.0$, label smoothing $\alpha = 0.1$
 1: Initialize step $k \leftarrow 0$, recent adversarial reward $\mathcal{Q} \leftarrow \emptyset$
 2: **while** training **do**
 3:      Sample melody $x \sim \mathcal{D}$
 4:      Generate online rollout $y \sim \pi_\theta(\cdot \mid x)$ using Eq. 2
 5:      Compute reward $R(x,y) = R_{\text{coh}}(x,y) + R_{\text{rules}}(x,y) + R_{\text{adv}}(x,y)$
 6:      Update policy $\pi_\theta$ with PPO on Eq. 4
 7:      Push $r_{\text{adv}}$ into queue $\mathcal{Q}$, keep last $m$ values
 8:      **if** SHOULDUPDATEDISC($k$, $\mathcal{Q}$, $T_{\text{warm}}$, $K_{\text{int}}$, $m$, $\tau$) **then**
 9:          DISCUPDATE($D_\psi$, $\mathcal{D}$, $\{y\}$, $\alpha$)                 ▷ see §3.3
10:      **end if**
11:      $k \leftarrow k + 1$
12: **end while**
13: **function** SHOULDUPDATEDISC($k$, $\mathcal{Q}$, $T_{\text{warm}}$, $K_{\text{int}}$, $m$, $\tau$)
14:      **if** $k \le T_{\text{warm}}$ **then**
15:          **return** ($k \bmod K_{\text{int}} = 0$)                  ▷ fixed interval during warmup
16:      **else**
17:          **return** $\left( |\mathcal{Q}|{=}m \right) \wedge \left( \frac{1}{m} \sum_{r \in \mathcal{Q}} r > \tau \right)$     ▷ confidence gated after warmup
18:      **end if**
19: **end function**

---

# A  USE OF LLM

We used LLMs to aid in writing and polishing this paper.

# B  APPENDIX

## B.1  ADDITIONAL RESULTS

## B.2  DATASET DETAILS

We use the Hooktheory dataset released in Donahue et al. (2022), which contains approximately 21,000 melody–chord pairs, and we follow its official train, validation, and test split. The POP909 dataset has 909 pairs, the Nottingham dataset has 1,019 pairs, and the Wikifonia dataset has 502 pairs.[2] For POP909 and Nottingham, we create random splits with 80% train, 10% validation, and 10% test.

For training the *chord accompaniment* policy, its offline baselines, and its reward models, we sample mini-batches from Hooktheory, POP909, and Nottingham with probabilities [60%, 30%, 10%]. For

---

[2] We use the public-domain Wikifonia subset from https://github.com/00sapo/OpenEWLD.

Table 1: Evaluation on model jamming with fixed melodies on the test set and the held-out test set. We report harmony quality (note-in-chord ratio) and diversity (Vendi Score); higher is better for both. Best system is **bold**, second best is underlined.

| System | Test set | | Out of distribution dataset | |
|---|---|---|---|---|
| | Harmony ↑ | Diversity ↑ | Harmony ↑ | Diversity ↑ |
| Online MLE | 0.368 | **29.491** | 0.362 | **16.401** |
| ReaLchords (Wu et al., 2024) | 0.484 | 20.968 | **0.475** | 8.417 |
| GAPT w/o Adv. Training | 0.476 | 20.814 | 0.447 | 8.034 |
| GAPT | **0.497** | 26.645 | 0.470 | 11.295 |
| Ground Truth | 0.727 | 27.922 | 0.784 | 10.962 |

Table 2: Evaluation on model jamming with a learned melody agent (model-to-model interaction) and in real-time user interaction. We report harmony quality (note-in-chord ratio) and diversity (Vendi Score); higher is better for both. Best system is **bold**, second best is underlined.

| System | Learned melody agent | | User interaction | |
|---|---|---|---|---|
| | Harmony ↑ | Diversity ↑ | Harmony ↑ | Diversity ↑ |
| Online MLE | **0.650** | **18.071** | 0.448 | **12.465** |
| ReaLchords (Wu et al., 2024) | 0.626 | 7.480 | 0.462 | 9.786 |
| GAPT w/o Adv. Training | 0.540 | 5.658 | N/A | N/A |
| GAPT | 0.648 | 12.914 | **0.467** | 11.794 |

training the *melody jamming* agent, its offline baselines, and its reward models, we sample from Hooktheory, POP909, Nottingham, and Wikifonia with probabilities [50%, 20%, 10%, 20%].

## B.3 EXPERIMENT DETAILS

The chord embedding of both the diversity and the t-SNE results are extracted from the chord encoder in the first full input contrastive reward models. The embeddings are already normalized at embedding extraction.

## B.4 USER STUDY DETAILS

The real-time interactive system generates music in chunks with a fixed lookahead of $t_f$ beats and a commit horizon of $t_c$ beats. The frontend maintains a buffer of planned outputs to handle network latency. For all user study sessions we set tempo to 80 BPM, sampling temperature to 0.8, $t_f = 4$ beats, $t_c = 4$ beats, and an initial listen only period of 8 beats before accompaniment begins. The frontend runs locally with a MIDI keyboard while the model serves from a remote GPU node.

For within-subject comparisons, each participant performs the same three tasks with each model: play the same melody (task one), apply the same key modulation (task two), and start the same co-improvisation prompt (task three). System parameters, including tempo and sampling temperature, are fixed across models.

We report paired $t$-tests with $p$-values for statistical significance in Fig. 3. Interaction histories have varied lengths and can exceed the maximum input length of the contrastive reward model. To compute diversity, we apply a sliding window with 50% overlap over each jamming session, embed each window, then aggregate diversity over all windows for a given model.

The user study conducted in this paper has received IRB approval. All participants signed a written informed consent form before any study activities began. Data were collected using secure, access controlled systems, encrypted in transit and at rest, and stored on institution managed servers. Identifiers were not retained beyond scheduling logistics, and all records were de-identified at ingestion by replacing names and contact information with randomly assigned participant codes. Access to the dataset was limited to the research team on a need to know basis, and any shared materials report

Table 3: The retrieval performance of contrastive reward models on the test set.

| Contrastive Reward Models | Note to Chord | | | | Chord to Note | | | |
|---|---|---|---|---|---|---|---|---|
| | R@1 | R@5 | R@10 | mAP@10 | R@1 | R@5 | R@10 | mAP@10 |
| model 1 | 0.18 | 0.41 | 0.54 | 0.28 | 0.18 | 0.41 | 0.53 | 0.28 |
| model 2 | 0.18 | 0.41 | 0.53 | 0.28 | 0.17 | 0.41 | 0.53 | 0.27 |
| rhythm-only model 1 | 0.04 | 0.14 | 0.21 | 0.08 | 0.05 | 0.13 | 0.20 | 0.08 |
| rhythm-only model 2 | 0.03 | 0.12 | 0.19 | 0.07 | 0.03 | 0.12 | 0.19 | 0.07 |

Table 4: The classification performance of discriminative reward models on the test set.

| Discriminative Reward Models | Precision | Recall | F1 |
|---|---|---|---|
| discriminative reward model 1 | 0.86 | 0.88 | 0.87 |
| discriminative reward model 2 | 0.87 | 0.87 | 0.87 |
| rhythm-only discriminative reward model 1 | 0.73 | 0.89 | 0.80 |
| rhythm-only discriminative reward model 2 | 0.77 | 0.82 | 0.79 |

only aggregate statistics or anonymized excerpts that cannot be linked to individuals. The study followed institutional policies and applicable privacy regulations throughout.

## C  MODEL ARCHITECTURE AND TRAINING DETAILS

### C.1  ARCHITECTURE AND PRE-TRAINING DETAILS OF ONLINE AND OFFLINE MODELS

**Online models.**  The online chord-accompaniment policy and the melody-jamming agent are decoder-only Transformers in the LLaMA-style family (Touvron et al., 2023). Each has 8 layers, 8 attention heads, and hidden dimension 512. We train with Adam (Kingma & Ba, 2015) at a fixed learning rate $1 \times 10^{-4}$ for 11,000 steps, batch size 64, and dropout rate 0.1. Positional encoding uses rotary position embeddings (RoPE) (Su et al., 2021).

**Offline models.**  The offline baselines for accompaniment and jamming use encoder–decoder Transformers with 8 encoder layers and 8 decoder layers, each with 8 attention heads and hidden dimension 512. We use relative position encodings following T5 (Raffel et al., 2020) instead of RoPE. Training uses Adam with learning rate $1 \times 10^{-4}$ for 13,000 steps, batch size 64, and dropout rate 0.1.

### C.2  TRAINING AND ARCHITECTURE DETAILS OF REWARD MODELS

**Contrastive reward model.**  Following Wu et al. (2024), we use a melody encoder and a chord encoder, each a 6-layer, 6-head Transformer encoder with hidden dimension 512. We $\ell_2$-normalize the outputs and train with the CLIP-style symmetric contrastive objective (Radford et al., 2021). We use Adam (Kingma & Ba, 2015) with learning rate $1 \times 10^{-4}$, batch size 196, and dropout rate 0.1. The full-input model is trained for 8,000 steps and the rhythm-only model for 2,500 steps. We train two instances per setting with different random seeds.

**Discriminative reward model.**  We use a 6-layer, 6-head Transformer encoder with hidden dimension 512. The input is the concatenation of melody and chords. We take the hidden state of the beginning-of-sequence token as the logit and train with binary cross-entropy using Adam (Kingma & Ba, 2015) at learning rate $1 \times 10^{-4}$ and dropout rate 0.1. The full-input model uses batch size 128 for 3,000 steps; the rhythm-only model uses batch size 64 for 3,000 steps to reduce overfitting.

### C.3  TEST-SET PERFORMANCE OF REWARD MODELS

See Tab. 3 for test set performance of contrastive reward models and Tab. 4 for test set performance of discriminative reward models.

Table 5: Simulated interaction results of the melody jamming agent on test set chords.

| System | Learned melody agent | |
|---|---|---|
| | Harmony ↑ | Diversity ↑ |
| Online MLE | 0.525 | 29.369 |
| Melody Jamming Agent | 0.654 | 28.124 |

## C.4 PERFORMANCE OF MELODY JAMMING AGENT

See Tab. 5 for harmony and diversity for the melody jamming agent compared with the melody online MLE model.

## C.5 RL POST-TRAINING DETAILS

We fine-tune the online accompaniment policy with PPO (Schulman et al., 2017) in an environment where the melody $x$ is given and the policy generates chords $y$. The total scalar reward for a rollout is

$$R(x, y) = R_{\text{coh}}(x, y) + R_{\text{rules}}(x, y) + R_{\text{adv}}(x, y), \tag{6}$$

where all three terms have equal weight. $R_{\text{coh}}$ is the contrastive coherence reward from the contrastive model, $R_{\text{rules}}$ is the rule-based musicality reward, and $r_{\text{adv}}$ is the adversarial reward derived from the discriminator. We treat the reward as trajectory-level for PPO updates.

**Optimization and schedules.** We run 1,000 PPO updates. Adam uses $\beta_1 = 0.9$ and $\beta_2 = 0.95$ for both actor and critic. Actor learning rate is $5 \times 10^{-7}$ and critic learning rate is $9 \times 10^{-6}$. We use linear warmup for the first 100 updates followed by cosine decay for the remaining 900 updates, with a floor at $10\%$ of the peak learning rate. Each PPO update uses batch size 384 and mini-batch size 48; we iterate over the 8 mini-batches per update. Entropy regularization uses coefficient $\gamma = 0.01$ in Eq. 4. The KL regularization uses coefficient $\beta = 0.001$ in Eq. 4. We use standard PPO defaults for gradient clipping and normalization.

## C.6 DISCRIMINATOR DETAILS

The discriminator $D_\psi$ is an 8-layer, 8-head Transformer encoder with hidden dimension 512. We take the hidden state of the beginning-of-sequence token as the classification logit. We train with Adam (Kingma & Ba, 2015) using $\beta_1 = 0.9$, $\beta_2 = 0.95$, dropout rate 0.1, linear warmup for 100 steps, and cosine decay for the next 900 steps. The peak learning rate is $9 \times 10^{-5}$ and the floor learning rate is $9 \times 10^{-6}$.

## C.7 HARMONY AND DIVERSITY ACROSS DIFFERENT RANDOM INITIALIZATION

Tables 6 report Harmony (note-in-chord ratio, %) and Diversity (Vendi Score) for GAPT, GAPT w/o Adv., and ReaLchords, each trained with 3 different random seeds for RL post training. For each system, we show the mean and standard deviation across seeds, and higher is better for both Harmony and Diversity.

Table 6: Harmony (note-in-chord ratio, %) and Diversity (Vendi Score) on the Test Set and held-out Wikifonia dataset across 3 different random seeds for RL post training. For each model, we report mean ± standard deviation across seeds. Higher is better for both metrics.

| Model | Test Set | | Held-out Dataset | |
|---|---|---|---|---|
| | Harmony ↑ | Diversity ↑ | Harmony ↑ | Diversity ↑ |
| GAPT (ours) | $0.497 \pm 0.017$ | $25.540 \pm 1.475$ | $0.470 \pm 0.014$ | $11.092 \pm 0.277$ |
| GAPT w/o Adv. | $0.477 \pm 0.001$ | $20.942 \pm 1.057$ | $0.445 \pm 0.002$ | $8.084 \pm 0.412$ |
| ReaLchords | $0.486 \pm 0.005$ | $21.431 \pm 0.780$ | $0.472 \pm 0.006$ | $8.519 \pm 0.253$ |

## C.8 TRAINING DYNAMICS OF GAPT

Figure 8 presents the training dynamics of GAPT, including (a) overall reward, (b) adversarial reward from the discriminator, (c) discriminator training accuracy, and (d) discriminator training loss over the course of RL post-training. The discriminator is updated for a total of 67 steps, with 40 updates in phase 1 (fixed-interval warmup) and 27 updates in phase 2 (adaptive updates). In our implementation, each discriminator update step aggregates 8 gradient descent updates, aligned with the 8 mini-batch gradient descent steps in PPO. The discriminator loss and accuracy reported at each step are therefore averages over 8 gradient updates, which makes the logged loss values lower and the logged accuracies higher than if they were recorded before each individual gradient update.

Figure 9 reports the average critic values predicted over the course of training for GAPT and for GAPT w/o Adv. The estimated values for GAPT are consistently slightly higher than those of GAPT w/o Adv. Figure 10 further shows the difference in estimated critic values between GAPT and GAPT w/o Adv. alongside the adversarial reward. The difference of critic value closely correspondence with the additional adversarial reward, indicating that the critic effectively estimates the additional contribution of the adversarial reward term.

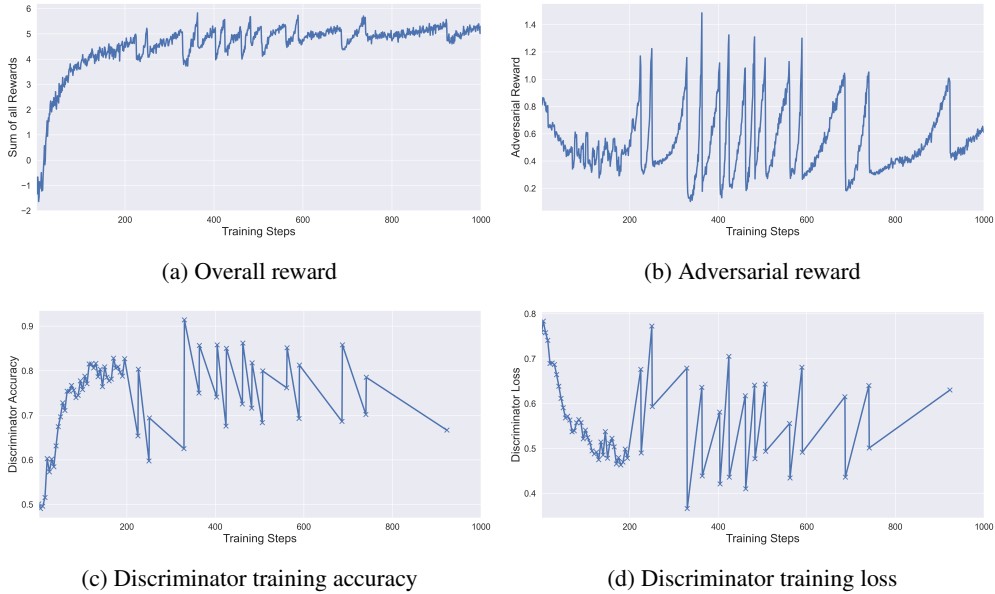

        (a) Overall reward                       (b) Adversarial reward

   (c) Discriminator training accuracy           (d) Discriminator training loss

Figure 8: Training dynamics of overall reward, adversarial reward, and discriminator performance during GAPT RL post-training.

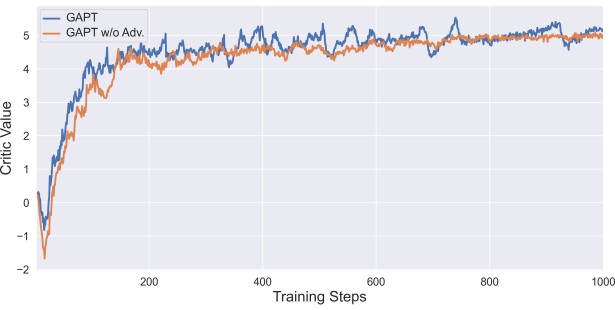

Figure 9: Estimated critic values for GAPT and GAPT w/o Adv. throughout the course of training.

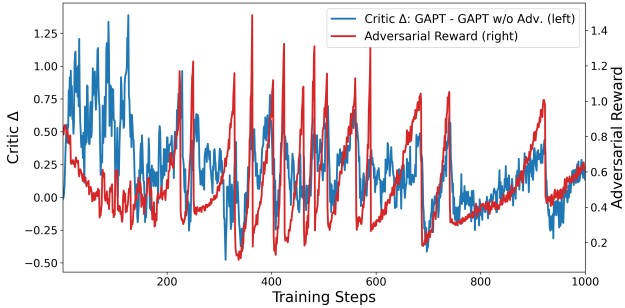

Figure 10: The difference of estimated critic values between GAPT and GAPT w/o Adv. compared with adversarial reward.

## C.9 ABLATION EXPERIMENTS ON REWARD WEIGHTING

To study the effect of the scalar reward in Equation 6, we vary the coefficients of the reward

$$R(x,y) \; = \; \alpha R_{\text{coh}}(x,y) \; + \; \beta R_{\text{rules}}(x,y) \; + \; \gamma R_{\text{adv}}(x,y), \tag{7}$$

where $R_{\text{coh}}(x,y)$ denotes the coherence reward, $R_{\text{rules}}(x,y)$ the sum of rule based penalties, and $R_{\text{adv}}(x,y)$ the adversarial realism reward. In the main experiments, we use $\alpha = \beta = \gamma = 1$. Here we keep the training setup and architecture fixed and only change the coefficients $(\alpha, \beta, \gamma)$. The results on both the test set and the held out Wikifonia dataset are reported in Table 7.

We first observe that upweighting the coherence term to $\alpha = 2, \beta = 1, \gamma = 1$ has very limited effect on both harmony and diversity, which indicates that GAPT is relatively robust to moderate changes of the coherence weight. In contrast, when we upweight the rule-based term ($\alpha = 1, \beta = 2, \gamma = 1$), both harmony and diversity drop on the test set and Wikifonia. This suggests that overemphasizing rule penalties discourages some harmonically reasonable but slightly rule violating chords, which reduces coverage and harms the balance between coherence and diversity. When we increase the adversarial weight ($\alpha = 1, \beta = 1, \gamma = 2$), harmony decreases more noticeably while diversity is also slightly reduced, showing that a too strong adversarial signal encourages the policy to chase discriminator preference at the expense of tight alignment with the input melody.

Removing the rule based rewards entirely ($\alpha = 1, \beta = 0, \gamma = 1$) leads to a clear form of reward hacking. The policy learns to exploit the remaining rewards by generating structurally invalid sequences, for example not following the onset plus holds pattern for each chord. In this case the outputs are degenerate and the harmony and diversity scores are not meaningful, which we indicate as N/A in the table. If we keep only the invalid sequence penalty $R_{\text{invalid}}$ inside $R_{\text{rules}}$ ($\alpha = 1, \beta = 0, \gamma = 1, +R_{\text{invalid}}$), the model again produces valid chord sequences and achieves reasonable harmony and diversity, but both metrics remain slightly worse than the full GAPT setting. Overall, this ablation shows that (i) the equal weighting $\alpha = \beta = \gamma = 1$ provides a good balance between coherence, adversarial realism, and diversity, and (ii) the rule based term, particularly beyond the invalidity penalty, plays an important role in preventing reward hacking and in stabilizing the tradeoff between harmonic accuracy and output diversity.

Table 7: Ablation study on reward component weighting in Equation 6. We report Harmony (note-in-chord ratio, %) and Diversity (Vendi Score) on the Test Set and Held-out Dataset (Wikifonia). Higher is better for both metrics.

| System | Test Set | | Held-out Dataset | |
|---|---|---|---|---|
| | Harmony ↑ | Diversity ↑ | Harmony ↑ | Diversity ↑ |
| GAPT | 0.497 | 26.645 | 0.470 | 11.295 |
| Upweight Coherence ($\alpha = 2, \beta = 1, \gamma = 1$) | 0.494 | 26.742 | 0.476 | 11.553 |
| Upweight Rules ($\alpha = 1, \beta = 2, \gamma = 1$) | 0.475 | 25.667 | 0.458 | 10.628 |
| Upweight Adversarial ($\alpha = 1, \beta = 1, \gamma = 2$) | 0.456 | 26.317 | 0.449 | 11.184 |
| Exclude Rules ($\alpha = 1, \beta = 0, \gamma = 1$) | N/A | N/A | N/A | N/A |
| Exclude Rules + Invalid Penalty ($\alpha = 1, \beta = 0, \gamma = 1, +R_{\text{invalid}}$) | 0.488 | 25.072 | 0.461 | 10.428 |

## C.10    Ablation Experiments on RL Objective

We conduct ablation experiments to evaluate the effect of adversarial training under different RL objectives, with results shown in Table 8. In addition to PPO (Schulman et al., 2017), we experiment with GRPO (Shao et al., 2024) for reward maximization. GRPO removes the critic model and instead generates multiple responses for the same input, defining the advantage as the reward normalized by the mean and standard deviation of rewards from responses to the same input. Specifically, we implement Dr.GRPO (Liu et al., 2025), a variant of GRPO that removes the standard deviation term in the normalization to obtain an unbiased estimator based on the mean reward.

As shown in Table 8, GRPO without adversarial training exhibits the same diversity collapse as PPO based RL post training. Adding adversarial training on top of GRPO restores diversity and slightly improves harmony on both the test set and the out of distribution dataset, mirroring the effect observed with GAPT. This supports that the benefit of adversarial training is robust across different RL objectives.

Table 8: Evaluation on model jamming with fixed melodies on the test set and the held out test set. We report harmony quality (note in chord ratio) and diversity (Vendi Score); higher is better for both.

| System | Test set | | Out of distribution dataset | |
| --- | --- | --- | --- | --- |
| | Harmony ↑ | Diversity ↑ | Harmony ↑ | Diversity ↑ |
| Online MLE | 0.368 | 29.491 | 0.362 | 16.401 |
| ReaLchords (Wu et al., 2024) | 0.484 | 20.968 | 0.475 | 8.417 |
| GAPT w/o Adv. Training | 0.476 | 20.814 | 0.447 | 8.034 |
| GAPT | 0.497 | 26.645 | 0.470 | 11.295 |
| GRPO w/o Adv. Training | 0.459 | 16.872 | 0.443 | 6.952 |
| GRPO w/ Adv. Training | 0.478 | 26.603 | 0.461 | 11.592 |
| Ground Truth | 0.727 | 27.922 | 0.784 | 10.962 |

## C.11    Ablation Experiments on Adversarial Discriminator Input

We further ablate the input to the adversarial discriminator by comparing a discriminator that takes only the chord trajectory as input, $D_\psi(y)$, with one that conditions on both melody and chord, $D_\psi(x, y)$. The results are reported in Table 9. When we train with $D_\psi(x, y)$, the model still achieves higher diversity than the setting without adversarial training, but its diversity remains lower than the original $D_\psi(y)$ variant, and harmony on the held out dataset also decreases slightly.

We hypothesize that conditioning the discriminator on both melody and chord makes it easier for $D_\psi$ to memorize specific training pairs, which reduces the effective difficulty of the discrimination task. This overfitting can weaken the adversarial signal and limit its ability to regularize the policy toward diverse yet realistic chord trajectories, thereby hurting generalization compared with the chord only discriminator.

Table 9: Evaluation on model jamming with fixed melodies on the test set and the held out test set. We report harmony quality (note in chord ratio) and diversity (Vendi Score); higher is better for both.

| System | Test set | | Out of distribution dataset | |
| --- | --- | --- | --- | --- |
| | Harmony ↑ | Diversity ↑ | Harmony ↑ | Diversity ↑ |
| Online MLE | 0.368 | 29.491 | 0.362 | 16.401 |
| ReaLchords (Wu et al., 2024) | 0.484 | 20.968 | 0.475 | 8.417 |
| GAPT w/o Adv. Training | 0.476 | 20.814 | 0.447 | 8.034 |
| GAPT | 0.497 | 26.645 | 0.470 | 11.295 |
| GAPT w/ $D_\psi(x, y)$ | 0.467 | 23.545 | 0.443 | 10.124 |
| Ground Truth | 0.727 | 27.922 | 0.784 | 10.962 |

## C.12    Average Number of Pitch Classes in Chords

Please refer to the following link for a full vocabulary of 2,821 distinct chord symbols, and this vocabulary is shared across all methods in the paper: `https://realchords-gapt.github.io/static/assets/chord_names_augmented.json`.

Table 10 presents average number of pitch classes in chord predicted by each model and ground truth. The averages are very close across the dataset and all models, which indicates that the models do not systematically exploit unusually dense chord symbols to inflate the metric.

Table 10: Average number of pitch classes in chords for different systems on the test set and the held out dataset (Wikifonia).

| System | Test set | Held out dataset |
|---|---|---|
| Dataset (Ground Truth) | 3.28 | 3.54 |
| Online MLE | 3.22 | 3.26 |
| ReaLchords | 3.01 | 3.00 |
| GAPT w/o Adv. | 3.00 | 3.00 |
| GAPT (ours) | 3.18 | 3.18 |

## C.13    Accompaniment Quality on Long Sequences

To evaluate model behavior beyond the RL horizon, we evaluate GAPT and the baselines on sequences of length $T = 512$ in two settings: (i) the held-out test split of the training corpora, (ii) the out-of-distribution Wikifonia dataset. We compare two inference strategies:

- **Naive extension.** We increase the transformer context length from 256 to 512 frames and roll out the policy autoregressively without changing the architecture or training procedure. We rely on the rotary positional embeddings (RoPE) used in our transformer and simply increase the maximum context length from 256 to 512 frames, which in principle allows positional extrapolation beyond the training horizon.

- **Sliding window.** We keep the maximum transformer context at 256 frames and maintain a moving window over the interaction history. When the sequence length exceeds 256 frames, the input window is incrementally shifted forward while we continue to generate online outputs. We use a hop size of 8 frames. Concretely, once the current sequence reaches $T = 256$ frames, we shift the context window by 8 frames to the right, predict the next 8 chord tokens in a batch while filling in the next 8 melody tokens as input, and repeat this procedure. This design keeps the effective context bounded while still exposing the model to a continuously updated history over long sessions.

Figure 11 reports results on 137 long sequences from the test set, and Figure 12 shows the same analysis on 300 long sequences from Wikifonia. In both datasets, the note-in-chord ratio remains stable across time, with no systematic degradation in harmonic coherence. Compared to the naive extension, the sliding window strategy obtains better harmony.

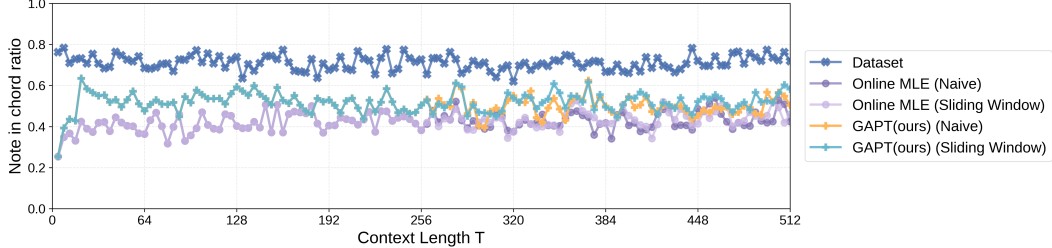

Figure 11: Note-in-chord ratio of GAPT on the test set for long sequences. We evaluate 137 sequences with total length $T \geq 512$ sixteenth-note frames and plot the average note-in-chord ratio as a function of time.

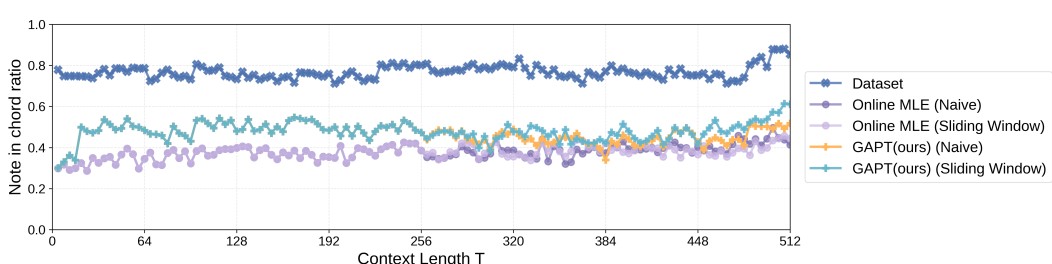

Figure 12: Note-in-chord ratio of `GAPT` on the held-out Wikifonia dataset for long sequences. We evaluate 300 sequences with total length $T \geq 512$ sixteenth-note frames and plot the average note-in-chord ratio as a function of time.

