# OpenReview forum: "Generative Adversarial Post-Training Mitigates Reward Hacking in Live Human-AI Music Interaction"
_ICLR.cc/2026/Conference — ICLR 2026 Poster_

### Official Review · Reviewer_wu95 · 2025-10-31

**Soundness:** 3
**Presentation:** 3
**Contribution:** 2
**Rating:** 6
**Confidence:** 4

**Summary:**

The paper proposes GAPT (Generative Adversarial Post-Training) to solve the problem of reward hacking in human-music interaction. The paper argues that current methods for RL post training in music related scenarios hacks the reward but leads to repetitive or low diversity music samples. To solve this, the authors add a discriminator component to distinguish between policy generated vs real-data trajectories. Since GAN based training can be unstable, they also propose training techniques to stabilize the training.

**Strengths:**

The paper addresses an important problem in live music jamming. The proposed solution of having an adversarial loss is simple and seems to work. Moreover, the authors have good evaluation setting, they evaluate on three scenarios: fixed-melody, model-model and real-time with musicians. The paper is very well written and easy to understand.

**Weaknesses:**

1. The scope of experimentation is narrow. For example, the proposed algorithm is tested on only melody-chord accompaniment. Similarly, the method is limited to T<=256 frames, where each frame is a sixteenth note. There are no studies in the paper that explains what happens for greater timesteps, is the training still stable?

2. The authors use note-in-chord ratio which is a very simple metric. While the qualitative results suggest that GAPT performs better, from Table 1 and 2, we see minimal improvement compared to other baselines. For instance, Table 2 shows that Online MLE performs better than GAPT in everything except for harmony in user interaction (0.448 vs 0.467 (GAPT)). The quantitative results are not convincing enough to prove the efficacy of this method.

**Questions:**

1. Could you provide empirical analysis of your method's performance and training stability for sequences longer than 256 frames? This is important because real musical interactions often extend beyond this length, and understanding any degradation in performance or training stability would significantly impact the method's practical utility.

2. In Table 2, Online MLE outperforms GAPT in most metrics except for harmony in user interaction. Could you explain this discrepancy and provide additional analysis to justify why GAPT should be preferred over the simpler Online MLE approach?

3. How does this method work for other musical styles?

---

> ### Author Response · Authors · 2025-11-21
>
> We thank reviewer wu95 for the careful reading and for emphasizing both the importance of broader scope and the need for convincing quantitative evidence. We appreciate your constructive questions and address them below.
>
> > The scope of experimentation is narrow. For example, the proposed algorithm is tested on only melody-chord accompaniment. Similarly, the method is limited to T<=256 frames, where each frame is a sixteenth note. There are no studies in the paper that explains what happens for greater timesteps, is the training still stable?
> > Could you provide empirical analysis of your method's performance and training stability for sequences longer than 256 frames? This is important because real musical interactions often extend beyond this length, and understanding any degradation in performance or training stability would significantly impact the method's practical utility.
>
> Thank you for these important concerns about scope and horizon length. We fully agree that understanding behavior beyond the training horizon is valuable for practical deployment.
>
> To evaluate model behavior beyond the RL horizon, we evaluate GAPT and the baselines on sequences of length $T = 512$ in two settings: (i) the held out test split of the training corpora, and (ii) the out of distribution Wikifonia dataset. We compare two inference strategies:
>
> - **Naive extension.** We increase the Transformer context length from 256 to 512 frames and roll out the policy autoregressively without changing the architecture or training procedure. We rely on the rotary positional embeddings (RoPE) used in our Transformer and simply increase the maximum context length from 256 to 512 frames, which in principle allows positional extrapolation beyond the training horizon.
> - **Sliding window.** We keep the maximum Transformer context at 256 frames and maintain a moving window over the interaction history. When the sequence length exceeds 256 frames, the input window is incrementally shifted forward while we continue to generate online outputs. We use a hop size of 8 frames. Concretely, once the current sequence reaches $T = 256$ frames, we shift the context window by 8 frames to the right, predict the next 8 chord tokens in a batch while filling in the next 8 melody tokens as input, and repeat this procedure. This design keeps the effective context bounded while still exposing the model to a continuously updated history over long sessions.
>
> **Figure 11** (line 1117) in the updated draft reports results on 137 long sequences from the test set, and **Figure 12** (line 1155) shows the same analysis on 300 long sequences from Wikifonia. In both datasets, **the note in chord ratio remains stable across time, with no systematic degradation in harmonic coherence, and the sliding window strategy consistently attains better harmony than the naive extension.** This indicates that our training remains stable and that GAPT maintains good performance even for sequences longer than the RL horizon.
>
> (cont.)

---

> ### Author Response · Authors · 2025-11-21
>
> (continued)
>
> > The authors use note-in-chord ratio which is a very simple metric. While the qualitative results suggest that GAPT performs better, from Table 1 and 2, we see minimal improvement compared to other baselines. For instance, Table 2 shows that Online MLE performs better than GAPT in everything except for harmony in user interaction (0.448 vs 0.467 (GAPT)). The quantitative results are not convincing enough to prove the efficacy of this method.
> > In Table 2, Online MLE outperforms GAPT in most metrics except for harmony in user interaction. Could you explain this discrepancy and provide additional analysis to justify why GAPT should be preferred over the simpler Online MLE approach?
>
> We thank the reviewer for this careful reading of **Table 1** and **Table 2** and for raising the important question of how to interpret the quantitative evidence.
>
> First, we fully agree that the note in chord ratio is a simple metric and not a complete measure of musical quality. It only captures whether the melody pitch class belongs to the concurrent chord, and cannot distinguish, for example, between rich but well voiced chords and trivial root position triads, or capture phrasing, tension and release, or longer term structure. We chose it as a transparent and widely used proxy for harmonic alignment, and we pair it with the Vendi diversity score to quantify coverage in the space of chord progressions.
>
> In the revised draft, **we add three further analyses in Appendix C**:
>
> 1. **Pitch class statistics in data versus model generations (Appendix C.12, Table 10, line 1085).** We report the average number of distinct pitch classes per chord for the ground truth annotations and for each model in **Appendix C.12, Table 10, line 1085**. Across both the test set and Wikifonia, all models have average pitch class counts that are very close to the dataset baseline and to each other, rather than systematically selecting chords with many more notes. **This indicates that none of the models, including GAPT, improves the note in chord ratio simply by predicting unusually dense chords; instead, they maintain a comparable level of chord complexity while improving alignment.**
>
> 2. **Harmony and diversity across three RL seeds (Appendix C.7, Table 6, line 910).** We train three independent RL runs for each of GAPT, GAPT without adversarial reward, and ReaLchords, and report the mean and standard deviation of note in chord ratio and Vendi diversity in **Appendix C.7, Table 6 (line 910)**. The standard deviations are small for all models, and **GAPT consistently achieves higher diversity than ReaLchords and the non adversarial ablation at similar or slightly better harmony.** This shows that the improvements are stable across seeds rather than due to random variation, and that the relative advantage of GAPT over coherence only RL is robust.
>
> 3. **GRPO as RL objectives (Appendix C.10, Table 8, line 1040).** In **Appendix C.10, Table 8 (line 1040)**, we compare PPO based RL post training with GRPO / Dr.GRPO. **GRPO without adversarial training exhibits the same diversity collapse as PPO based RL post training, while adding adversarial training on top of GRPO restores diversity and slightly improves harmony on both the test set and the out of distribution dataset.** This confirms that **the benefit of the adversarial reward holds across different RL objectives**, reinforcing that GAPT’s gains are not specific to a single policy gradient algorithm.
>
> Regarding the specific question about **Table 2** and Online MLE, we agree that at first glance it can seem surprising that Online MLE performs strongly in several entries. The key point is that **Table 2** corresponds to co adaptive settings, which differ qualitatively from the fixed melody setting of **Table 1**:
>
> - In the learned melody agent setting (left half of **Table 2**), the melody agent is explicitly trained to adapt to the chord policy. This is a favorable scenario for Online MLE, which tends to play more static progressions and is less reactive to incoming melody. When its partner is trained to follow those chords, the joint trajectory stays close to the supervised training distribution, and both harmony and diversity metrics can look strong even though the chord policy itself is not very adaptive.
> - In the user interaction setting (right half of **Table 2**), expert musicians are also able to adjust their playing to the chords they hear, which can partially compensate for the limitations of simpler policies. This co adaptation again helps Online MLE, and we see that its note in chord ratio is close to that of GAPT. At the same time, **GAPT achieves the highest harmony in user interaction (0.467 versus 0.448 for Online MLE) while maintaining comparable diversity, placing it on the empirical Pareto frontier of harmony and diversity for human sessions.**
>
> (cont.)

---

> > ### Author Response · Authors · 2025-11-21
> >
> > (continued)
> >
> > Because our target application is live jamming with human musicians, we view the human preference results as crucial complementary evidence. **Figure 3** shows that GAPT attains the highest mean rating on all three questions, and significantly improves perceived adaptation speed and perceived control and agency compared with ReaLchords. Qualitative comments also highlight that participants felt GAPT “catches my key and chord changes faster” and “adapts faster,” whereas Online MLE was described as sometimes producing mismatched chords even when it occasionally explored interesting progressions.
> >
> > **Taken together, we therefore see the picture as follows:** note in chord ratio is an imperfect but useful harmonic proxy, which we now complement with pitch class statistics to rule out trivial chord inflation, and three seed statistics to establish robustness. Within this richer set of quantitative checks, **GAPT reliably improves the tradeoff between harmony and diversity over coherence only RL, and matches or slightly improves harmonic alignment relative to Online MLE in human sessions. Crucially, these differences are aligned with the user study, where expert musicians systematically prefer GAPT in terms of adaptation and perceived agency, which we believe is the most relevant criterion for this interactive setting.**
> >
> > > How does this method work for other musical styles?
> >
> > Thank you very much for this thoughtful question about stylistic coverage. We fully agree that evaluating and extending the method to a broader range of musical styles is an important direction, and we would be excited to explore this in future work.
> >
> > Our current datasets (Hooktheory, POP909, Nottingham, and Wikifonia) predominantly consist of Western popular and folk music, so both the models and the reward functions are tuned to that setting. In principle, however, **the GAPT framework itself is style agnostic**: as long as we can define appropriate reward models (for example, style specific coherence models and musical rules) and collect data for the target style, the same adversarial post training procedure can be applied.
> >
> > At the same time, we expect that the underlying representation and base architecture may need to be adapted for some genres. For instance, in jazz and many other styles, expressive timing, swing, and triplets are common, so quantizing everything into sixteenth note frames is not ideal. In such cases, one could move to more flexible symbolic representations, for example event based encodings as in Music Transformer [1], and redesign the relevant rule based components accordingly. **We would be excited to investigate these representation and reward adaptations for jazz and other rich musical traditions in future work.**
> >
> > [1] Huang, C. A., Vaswani, A., Uszkoreit, J., Shazeer, N., Hawthorne, C., Dai, A. M., Hoffman, M. D., & Eck, D. (2018). Music Transformer: Generating music with long term structure. In Proceedings of ICLR.

---

### Official Review · Reviewer_VfzC · 2025-11-01

**Soundness:** 3
**Presentation:** 4
**Contribution:** 3
**Rating:** 6
**Confidence:** 3

**Summary:**

The paper proposed Generative Adversarial Post-Training (GAPT) for live melody→chord accompaniment. A discriminator is trained to prevent reward hacking, ensuring the realism of the accompaniment. The experiment results show that with GAPT, the model outputs more diverse content.

**Strengths:**

1. Promise to release training datasets, model checkpoints, and code.
2. Well written, well structured, clear problem and motivation.
3. Comprehensive experiments show the improvement of diversity, harmonic coherence, adaptation speed and user agency.

**Weaknesses:**

1. The chord set of the model is not specified; note-in-chord could be inflated by choosing chords with more pitch classes.
2. The authors did not benchmark against other RL post-training methods.

**Questions:**

1. What if the discriminator conditioned on melody D(x,y)?
2. What’s the latency budget and end-to-end timing under live use? Does GAPT change compute/latency vs RealChords?
3. In the appendix, you mentioned that for all user study sessions, you set the tempo to 80 BPM. Does that mean when participants are improvising, the bpm should be set to 80?

---

> ### Author Response · Authors · 2025-11-21
>
> We sincerely thank reviewer VfzC for the very positive and detailed review, and for highlighting the strengths in writing, structure, and evaluation. We are especially grateful for your comments on releasing datasets, models, and code, and for the concrete questions that helped us clarify important details.
>
> > The chord set of the model is not specified; note-in-chord could be inflated by choosing chords with more pitch classes.
>
> Thank you for pointing this out and for raising the concern about potential inflation of the note in chord ratio.
>
> In the revision, we now explicitly specify the chord vocabulary used by all models (**Section C.12, line 1080**). We provide the full list of chord symbols at
> https://realchords-gapt.github.io/static/assets/chord_names_augmented.json. The accompaniment policy predicts over a fixed vocabulary of **2,821 distinct chord symbols**, and this vocabulary is shared across all methods in the paper.
>
> Each symbol is factored into a root pitch class, a chord quality (triads and sevenths), optional extensions, and an optional inversion bass. Concretely, the vocabulary includes:
>
> - Major, minor, diminished, and augmented triads.
> - Dominant, major, and minor seventh chords with common extensions (for example 9, 11, 13).
> - Chords with added tones and alterations (for example add2, add4, add6, b13).
> - Suspended and pedal chords.
> - Standard slash chords for inversions.
>
> To directly address the concern that the note in chord ratio might be artificially increased by predicting excessively dense chords, we compute the **average number of distinct pitch classes per chord** for the ground truth annotations and for each model’s predictions. **The results are summarized in Table 10 of Appendix C.12 and in the table below.** **The averages of pitch classes from model predictions (3.00–3.22) are very close to the dataset averages (3.28–3.54), which indicates that the models do not systematically exploit unusually dense chord symbols to inflate the metric.** We add this table and explicitly reference it in **Section C.12** in the updated paper.
>
> **Table 10: Average number of pitch classes in chords for different systems on the test set and the held out dataset (Wikifonia).**
>
> | System                 | Test set | Held out dataset |
> | ---------------------- | -------- | ---------------- |
> | Dataset (Ground Truth) | 3.28     | 3.54             |
> | Online MLE             | 3.22     | 3.26             |
> | ReaLchords             | 3.01     | 3.00             |
> | GAPT w/o Adv.          | 3.00     | 3.00             |
> | GAPT (ours)            | 3.18     | 3.18             |
>
>
>
>
>
> > The authors did not benchmark against other RL post-training methods.
>
> Thank you for this suggestion. We agree that comparing to other RL post training objectives is important.
>
> As mentioned in our response to reviewer SxDd, we conduct additional experiments using GRPO [1] instead of PPO, and more specifically the Dr. GRPO variant [2] that omits division by the standard deviation during normalization. Due to GPU memory constraints, we train GRPO with batch size 192, $n = 8$ rollouts per input, and batch size 192 for learning updates. **We include the result of GRPO in Table 8 (line 1040) of the updated draft, and the table below.** **GRPO without adversarial training exhibits the same diversity collapse as PPO based RL post training, while adding adversarial training on top of GRPO restores diversity and slightly improves harmony on both the test set and the out of distribution dataset, mirroring the effect observed with GAPT.** This supports that **the benefit of adversarial training is robust across different RL objectives**.
>
> We will include these comparisons in the revised paper and clarify that **the core contribution of GAPT, namely the adversarial reward and adaptive discriminator schedule, is compatible with different underlying policy gradient algorithms.**
>
> (cont.)

---

> > ### Author Response · Authors · 2025-11-21
> >
> > (continued)
> >
> > **Table 8: Evaluation on model jamming with fixed melodies on the test set and the held out test set. We report harmony quality (note in chord ratio) and diversity (Vendi Score); higher is better for both.**
> >
> > | System                 | Test set Harmony ↑ | Test set Diversity ↑ | Out of distribution dataset Harmony ↑ | Out of distribution dataset Diversity ↑ |
> > | ---------------------- | ------------------ | -------------------- | ------------------------------------- | --------------------------------------- |
> > | Online MLE             | 0.368              | 29.491               | 0.362                                 | 16.401                                  |
> > | ReaLchords             | 0.484              | 20.968               | 0.475                                 | 8.417                                   |
> > | GAPT w/o Adv. Training | 0.476              | 20.814               | 0.447                                 | 8.034                                   |
> > | GAPT                   | 0.497              | 26.645               | 0.470                                 | 11.295                                  |
> > | GRPO w/o Adv. Training | 0.459              | 16.872               | 0.443                                 | 6.952                                   |
> > | GRPO w/ Adv. Training  | 0.478              | 26.603               | 0.461                                 | 11.592                                  |
> > | Ground Truth           | 0.727              | 27.922               | 0.784                                 | 10.962                                  |
> >
> > [1] Shao, Z., Wang, P., Zhu, Q., Xu, R., Song, J., Bi, X., ... & Guo, D. (2024). Deepseekmath: Pushing the limits of mathematical reasoning in open language models. arXiv preprint arXiv:2402.03300.
> > [2] Liu, Z., Chen, C., Li, W., Qi, P., Pang, T., Du, C., ... & Lin, M. (2025). Understanding r1-zero-like training: A critical perspective. arXiv preprint arXiv:2503.20783.
> >
> >
> >
> > > What if the discriminator conditioned on melody D(x,y)?
> >
> > We appreciate this interesting suggestion about the discriminator architecture.
> >
> > We run an additional experiment where we modify GAPT so that the discriminator takes both the input melody and the generated accompaniment as input, that is $D_{\psi}(x, y)$, and learns to judge whether the joint pair comes from data or from the current policy. In this version, the adversarial reward becomes a function of both melody and chords. The results are reported in **Table 9** of the updated draft **(line 1067)** and in the table below. When we train with $D_{\psi}(x, y)$, the model still achieves higher diversity than the setting without adversarial training, but its diversity remains lower than the original $D_{\psi}(y)$ variant, and harmony on the held out dataset also decreases slightly.
> >
> > **We hypothesize that conditioning the discriminator on both melody and chord makes it easier for $D_{\psi}$ to memorize specific training pairs, which reduces the effective difficulty of the discrimination task.** This overfitting can weaken the adversarial signal and limit its ability to regularize the policy toward diverse yet realistic chord trajectories, thereby hurting generalization compared with the chord only discriminator.
> >
> > **Table 9: Evaluation on model jamming with fixed melodies on the test set and the held out test set. We report harmony quality (note in chord ratio) and diversity (Vendi Score); higher is better for both.**
> >
> > | System                   | Test set Harmony ↑ | Test set Diversity ↑ | Out of distribution dataset Harmony ↑ | Out of distribution dataset Diversity ↑ |
> > | ------------------------ | ------------------ | -------------------- | ------------------------------------- | --------------------------------------- |
> > | Online MLE               | 0.368              | 29.491               | 0.362                                 | 16.401                                  |
> > | ReaLchords               | 0.484              | 20.968               | 0.475                                 | 8.417                                   |
> > | GAPT w/o Adv. Training   | 0.476              | 20.814               | 0.447                                 | 8.034                                   |
> > | GAPT                     | 0.497              | 26.645               | 0.470                                 | 11.295                                  |
> > | GAPT w/ $D_{\psi}(x, y)$ | 0.467              | 23.545               | 0.443                                 | 10.124                                  |
> > | Ground Truth             | 0.727              | 27.922               | 0.784                                 | 10.962                                  |
> >
> > (cont.)

---

> > > ### Author Response · Authors · 2025-11-21
> > >
> > > (continued)
> > >
> > >
> > >
> > > > What’s the latency budget and end-to-end timing under live use? Does GAPT change compute/latency vs RealChords?
> > >
> > > Thank you for raising this practical question.
> > >
> > > The end to end timing for live use is:
> > >
> > > - **Backend compute.** Around 200 to 300 ms per request on an L40S GPU, depending on the requested generation horizon.
> > > - **Network.** Less than 100 ms round trip in our user study environment.
> > >
> > > **GAPT does not change the model architecture or inference time at deployment.** It is purely a different post training procedure. All models compared in the user study (Online MLE, ReaLchords, and GAPT) use the same 8 layer Transformer architecture with identical FLOPs. The discriminator is not used at inference time, so **there is no additional latency overhead for GAPT compared to ReaLchords.**
> > >
> > > > In the appendix, you mentioned that for all user study sessions, you set the tempo to 80 BPM. Does that mean when participants are improvising, the bpm should be set to 80?
> > >
> > > Yes, that is correct. Thank you for asking for clarification.
> > >
> > > In all user study sessions, we fix the tempo to 80 BPM, including during improvisation. The system also plays a metronome click at 80 BPM so that participants have a stable tempo reference while improvising. This choice controls for confounding effects of tempo variability on participants’ subjective impressions. **However, varying the tempo does not change the model’s behavior, since the data representation is defined in relative time on a grid of sixteenth note frames.**

---

> > > > ### Comment · Reviewer_VfzC · 2025-11-25
> > > >
> > > > My questions are fully answered. I raised my score.

---

### Official Review · Reviewer_SxDd · 2025-11-01

**Soundness:** 3
**Presentation:** 3
**Contribution:** 3
**Rating:** 6
**Confidence:** 4

**Summary:**

This work explores the use of adversarial learning to mitigate the reward hacking problem in reinforcement learning for the task of melody-to-chord accompaniment generation.
Reward hacking in RL often arises because (1) human-defined rewards are incomplete and may lead the policy to converge to a local minimum, and (2) RL agents cannot distinguish out-of-distribution (OOD) samples, causing them to over-rely on the reward signal.

What makes this paper interesting is how it integrates adversarial learning into the RL framework. Unlike other methods (e.g., [1]), the authors adopt a simple yet direct strategy: they add the discriminator term −log(1−y) directly to the reward. In this way, if the discriminator considers a generated sample to be “real,” the policy receives an additional (and potentially large) reward boost.

Experimentally, the authors demonstrate strong results in both quantitative metrics and perceptual evaluations. As an amateur music enthusiast, I can clearly perceive the improvement brought by the adversarial reward.

[1] Bukharin et al., Adversarial Training of Reward Models, 2025.

**Strengths:**

**Technical Comments**

1. Exploring adversarial learning in the context of melody-to-chord accompaniment is quite novel. Moreover, directly integrating the discriminator’s output into the reward function is a simple yet effective strategy, as demonstrated in the paper.

2. The authors present several insightful analyses and results. For example, in Figure 4, they show that the adversarial reward encourages more diverse generation outputs and expands the representational space. This finding is particularly interesting, since—according to the authors—other components such as entropy regularization remain unchanged compared to the w/o-adversarial baseline.

**Presentation**

The writing is clear and easy to follow, and the examples used in the comparative experiments are also quite convincing when listened to.

**Experiments**

1. For the main task, the authors demonstrate significant improvements over both the baseline and the w/o-adversarial-reward setting, in terms of quantitative metrics and subjective evaluation.
2. The ablation study is also comprehensive and effectively supports the paper’s claims.

**Weaknesses:**

**Technical Comments**

1. Since the adversarial reward takes values in  [0,+∞) and is directly added to the original reward, does this cause any numerical instability during training? Has the author conducted experiments to verify its stability?

2. Adversarial training (GAN) itself is known to be unstable. Did the author perform repeatability experiments to show the variance across different random initializations?

3. What does the reward convergence curve look like during training? It seems that this information is not presented in the paper.

4. Could the author show how the critic value differs between the models with and without adversarial reward? (I notice the authors use PPO instead of GRPO.)
Additionally, could they run an experiment using GRPO, since GRPO normalizes rewards using the group/batch average?
Given that the adversarial reward is always positive, it might shift the overall value distribution upward, potentially affecting learning stability and normalization.

5. Is the discriminator’s training process stable? What does its learning curve look like over the course of training?


I am very interested in this paper and I genuinely like the direction. My current score mainly reflects the fact that I would like to see a clearer and more complete picture from the authors, so that the contribution becomes more solid. If the authors can provide more thorough/deep analysis in the rebuttal or revision, I would be happy to raise my score.

**Questions:**

see above

---

> ### Author Response · Authors · 2025-11-21
>
> We are very grateful to reviewer SxDd for the thoughtful and enthusiastic review, and for explicitly encouraging us to provide a clearer picture of stability and variance. We appreciate your interest in the direction and your careful reading of the technical details. Below we respond to each of your technical questions. We reorganize the order of the questions for better response.
>
> > Adversarial training (GAN) itself is known to be unstable. Did the author perform repeatability experiments to show the variance across different random initializations?
>
> Thank you for highlighting this important concern. To assess repeatability, we conduct additional experiments where we train two additional random seeds for RL for each of GAPT, GAPT w/o Adv., and ReaLchords, resulting in three seeds per model. We then evaluate on the test set and report the mean and standard deviation of note in chord ratio and diversity score across these three seeds. **The results are summarized in Table 6 (line 909) of the revised manuscript and in the table below.**
>
> **As shown in Table, the standard deviations of both metrics are small for all three models, including GAPT, which indicates that our RL post training, together with the two phase discriminator schedule, yields low variance across random initializations.** We will make this explicit in the revised version.
>
>
>
> | Model         | Test Set Harmony $\uparrow$ | Test Set Diversity $\uparrow$ | Held-out Dataset Harmony $\uparrow$ | Held-out Dataset Diversity $\uparrow$ |
> | ------------- | --------------------------- | ----------------------------- | ----------------------------------- | ------------------------------------- |
> | GAPT (ours)   | $0.497 \pm 0.017$           | $25.540 \pm 1.475$            | $0.470 \pm 0.014$                   | $11.092 \pm 0.277$                    |
> | GAPT w/o Adv. | $0.477 \pm 0.001$           | $20.942 \pm 1.057$            | $0.445 \pm 0.002$                   | $8.084 \pm 0.412$                     |
> | ReaLchords    | $0.486 \pm 0.005$           | $21.431 \pm 0.780$            | $0.472 \pm 0.006$                   | $8.519 \pm 0.253$                     |
>
>
>
> > Since the adversarial reward takes values in [0,+∞) and is directly added to the original reward, does this cause any numerical instability during training? Has the author conducted experiments to verify its stability?
> >
> > Is the discriminator’s training process stable? What does its learning curve look like over the course of training?
> >
> > What does the reward convergence curve look like during training? It seems that this information is not presented in the paper.
>
> We appreciate these detailed questions about stability. In the revised draft (**Figure 8**, line 938), we now present the training dynamics for GAPT, including
> (a) overall scalar reward,
> (b) adversarial reward from the discriminator,
> (c) discriminator training accuracy, and
> (d) discriminator training loss
> as functions of RL update steps (**i.e. we now include the learning curves you requested**). Over training, the discriminator is updated for 67 steps in total, with 40 updates in phase 1 (fixed interval warmup) and 27 updates in phase 2 (adaptive updates). In our implementation, each discriminator “step” aggregates 8 gradient descent updates, aligned with the 8 mini batch PPO updates, and we log the loss and accuracy averaged over these 8 updates. **This aggregation explains why the logged discriminator loss appears relatively low and the accuracies relatively high in the learning curves.**
>
> Regarding the adversarial reward value: although in principle $R_{\text{adv}} \in [0, +\infty)$, in practice its value remains moderate throughout training:
>
> - After the warmup phase, where the discriminator is updated every 5 PPO steps, the discriminator quickly reaches reasonably high accuracy, which corresponds to a relatively low starting reward.
> - In the second phase with adaptive updates, the adversarial reward slowly increases as the policy improves. We trigger a discriminator update whenever the moving average of the last three adversarial rewards exceeds 1.0. After each such update, the reward drops sharply, then gradually ramps up again.
>
> **This schedule prevents the adversarial term from growing without bound and keeps it in a numerically stable range.** The corresponding reward and learning curves in **Figure 8** empirically illustrate that both the discriminator and the overall reward evolve smoothly rather than exhibiting outlier behavior. **Together with the low variance across seeds (Table 6), these results support that our adversarial post training is stable in practice.**
>
> (cont.)

---

> > ### Author Response · Authors · 2025-11-21
> >
> > (continued)
> >
> > > Could the author show how the critic value differs between the models with and without adversarial reward? (I notice the authors use PPO instead of GRPO.) Additionally, could they run an experiment using GRPO, since GRPO normalizes rewards using the group/batch average? Given that the adversarial reward is always positive, it might shift the overall value distribution upward, potentially affecting learning stability and normalization.
> >
> > Thank you for this suggestion. We agree that comparing critic values and testing GRPO provides useful additional insight.
> >
> > **Critic value comparison.**
> > We now report the average critic values over the course of training for both GAPT and GAPT w/o Adv. in **Figure 9** of the updated draft **(line 961)**. As expected, the critic values for GAPT are slightly higher due to the extra positive adversarial reward added on top of the coherence and rule based terms. We further show the difference in estimated critic values between GAPT and GAPT w/o Adv. alongside the adversarial reward in **Figure 10** of the updated draft **(line 972)**. **The difference in critic values closely corresponds to the additional adversarial reward, indicating that the critic effectively estimates the incremental contribution of the adversarial reward term.**
> >
> > **GRPO experiment.**
> > We also conduct experiments with GRPO [1] instead of PPO as the RL objective. Specifically, we implement Dr. GRPO [2], which normalizes rewards by the group mean but does not divide by the standard deviation, thereby avoiding bias. Due to memory constraints, we run GRPO with batch size 192, $n = 8$ rollouts per input, and batch size 192 for updates (so 8 updates per rollout step). We include the result of GRPO in **Table 8** (line 1040) of the updated draft. **GRPO without adversarial training exhibits the same diversity collapse as PPO based RL post training, while adding adversarial training on top of GRPO restores diversity and slightly improves harmony on both the test set and the out of distribution dataset, mirroring the effect observed with GAPT.** This supports that the benefit of adversarial training is robust across different RL objectives.
> >
> > **Overall, these comparisons in Figure 9, Figure 10, and Table 8 show that the adversarial reward shifts the value function in a controlled way and that the advantages of GAPT are consistent across PPO and GRPO style training.** We will include these comparisons in the revised paper and clarify that the core contribution of GAPT, namely the adversarial reward and adaptive discriminator schedule, is compatible with different underlying policy gradient algorithms.
> >
> > [1] Shao, Z., Wang, P., Zhu, Q., Xu, R., Song, J., Bi, X., … & Guo, D. (2024). Deepseekmath: Pushing the limits of mathematical reasoning in open language models. arXiv preprint arXiv:2402.03300.
> > [2] Liu, Z., Chen, C., Li, W., Qi, P., Pang, T., Du, C., … & Lin, M. (2025). Understanding r1 zero like training: A critical perspective. arXiv preprint arXiv:2503.20783.
> >
> > ---
> >
> > **Table 8: Evaluation on model jamming with fixed melodies on the test set and the held out test set. We report harmony quality (note in chord ratio) and diversity (Vendi Score); higher is better for both.**
> >
> > | System                 | Test set Harmony ↑ | Test set Diversity ↑ | Out of distribution dataset Harmony ↑ | Out of distribution dataset Diversity ↑ |
> > | ---------------------- | ------------------ | -------------------- | ------------------------------------- | --------------------------------------- |
> > | Online MLE             | 0.368              | 29.491               | 0.362                                 | 16.401                                  |
> > | ReaLchords             | 0.484              | 20.968               | 0.475                                 | 8.417                                   |
> > | GAPT w/o Adv. Training | 0.476              | 20.814               | 0.447                                 | 8.034                                   |
> > | GAPT                   | 0.497              | 26.645               | 0.470                                 | 11.295                                  |
> > | GRPO w/o Adv. Training | 0.459              | 16.872               | 0.443                                 | 6.952                                   |
> > | GRPO w/ Adv. Training  | 0.478              | 26.603               | 0.461                                 | 11.592                                  |
> > | Ground Truth           | 0.727              | 27.922               | 0.784                                 | 10.962                                  |

---

> > > ### Comment · Reviewer_SxDd · 2025-11-24
> > >
> > > I appreciate authors hard working on additional experiments and revision. In my view, the picture of the proposed Adv+RL training method is much clearer now. I raised my score.

---

### Official Review · Reviewer_pmqm · 2025-11-01

**Soundness:** 3
**Presentation:** 3
**Contribution:** 3
**Rating:** 6
**Confidence:** 3

**Summary:**

The paper proposes Generative Adversarial Post-Training (GAPT) for real-time melody-to-chord accompaniment. GAPT introduces a discriminator to avoid reward hacking in RL finetuning. Evaluation shows this simple yet effective GAPT framework works to mitigate reward hacking.

**Strengths:**

1. Evaluation metrics are reasonable. Note-in-chord ratio allows flexibility because of the nature of real-time accompaniment, but at the same time evaluates the quality of chords.
2. Adversarial realism reward is easy to add and conceptually orthogonal to KL/entropy regularization. The confidence-gated update is a neat stability trick.
3. The RL post training scalar reward not only include discriminative models, but also include music rules which mitigate the bias brought by discriminative model as well.
4. Appreciate the examples provided to have a straightforward evaluation in ears.

**Weaknesses:**

1. I would appreciate that if more ablations study would be completed on GAPT itself, especially on the scalar reward in Equation (6), applying different weights to see the behavior of models would be a helpful study to provide more insights.

**Questions:**

1. What could be the reason for harmonic incoherence (like in Hooktheory Example 4)? Is it because the melody has higher density of notes in beats comparing to other successful accompaniments? When GAPT underperforms harmony (e.g., slight drop on OOD), what musical patterns cause it? Maybe because of the training dataset?
2. What are end-to-end latency numbers (ms) and GPU budget during live use? How does lookahead/commit horizon trade off with perceived responsiveness?
3. Would you mind elaborating a bit more on applying GAPT outside real-time accompaniment?

---

> ### Author Response · Authors · 2025-11-21
>
> We sincerely thank reviewer pmqm for the detailed and constructive review, and for the many helpful suggestions that helped us improve the paper. Below we address your questions in turn.
>
> > I would appreciate that if more ablations study would be completed on GAPT itself, especially on the scalar reward in Equation (6), applying different weights to see the behavior of models would be a helpful study to provide more insights.
>
> We thank the reviewer for this very helpful suggestion. We fully agree that understanding how the scalar reward in Equation (6) behaves under different weightings is important for interpreting GAPT and for assessing the robustness of our design.
>
> Recall that our scalar reward is defined as
>
> $$
> R(x,y) \;=\; \alpha R_{\text{coh}}(x,y) \;+\; \beta R_{\text{rules}}(x,y) \;+\; \gamma R_{\text{adv}}(x,y),
> $$
>
> where $R_{\text{coh}}(x,y)$ is the coherence reward, $R_{\text{rules}}(x,y)$ is the sum of rule-based penalties, and $R_{\text{adv}}(x,y)$ is the adversarial realism reward from the discriminator. In the main experiments we use $\alpha = \beta = \gamma = 1$. To study the effect of the scalar combination, we perform an additional ablation in which we keep the training setup, architecture, and optimization scheme fixed and vary only the coefficients $\alpha, \beta, \gamma$:
>
> - Upweight coherence: $\alpha = 2, \beta = 1, \gamma = 1$
> - Upweight rules: $\alpha = 1, \beta = 2, \gamma = 1$
> - Upweight adversarial: $\alpha = 1, \beta = 1, \gamma = 2$
> - Exclude rules: $\alpha = 1, \beta = 0, \gamma = 1$
> - Exclude rules but keep invalid penalty: $\alpha = 1, \beta = 0, \gamma = 1$ with an additional $R_{\text{invalid}}$ term that penalizes structurally invalid chord sequences (for example not following the onset plus holds pattern for each chord).
>
> We report Harmony (note-in-chord ratio) and Diversity (Vendi score) on both the test set and the held-out Wikifonia dataset in **Table 7** of the revised manuscript in Section C.9 (line 1015) and in the **table below**. The main findings are:
>
> **Sensitivity of GAPT to moderate reweighting.** When we double the coherence weight $(\alpha = 2, \beta = 1, \gamma = 1)$, the results remain very close to the original GAPT setting: harmony and diversity change only slightly on both the test set and Wikifonia. **This suggests that GAPT is relatively robust to moderate upweighting of the coherence term.**
>
> **Overemphasizing rule-based penalties slightly hurts both metrics.** When we double the rule-based weight $(\alpha = 1, \beta = 2, \gamma = 1)$, both harmony and diversity decrease compared to the original setting. Intuitively, a strong rule penalty discourages some harmonically reasonable but slightly rule-violating chords, which reduces coverage and diversity while not providing clear gains in note-in-chord ratio. **This indicates that overly strong rule-based penalties can reduce both harmony and diversity.**
>
> **Overemphasizing adversarial reward trades off harmony for realism.** When we double the adversarial weight $(\alpha = 1, \beta = 1, \gamma = 2)$, we observe a noticeable drop in harmony and a small decrease in diversity. This indicates that if the discriminator term dominates too strongly, the policy chases trajectories that look realistic under the discriminator but are less tightly aligned with the input melody, confirming that the adversarial term must be balanced with coherence. **In other words, an overpowered adversarial term improves realism at the cost of harmonic alignment.**
>
> **Removing rule rewards leads to reward hacking through invalid outputs.** When we remove the rule-based reward entirely $(\alpha = 1, \beta = 0, \gamma = 1)$, the policy quickly learns to exploit the remaining reward signals by generating structurally invalid sequences (for example omitting required onset tokens or mixing inconsistent hold tokens). In this case the model collapses to invalid outputs and standard harmony and diversity metrics are not meaningful, which we indicate as N/A in **Table 7**. **This provides direct evidence that rule-based constraints are essential to prevent a particularly degenerate form of reward hacking.**
>
> **Keeping only the invalid penalty partially stabilizes training.** When we retain only the invalid output penalty $R_{\text{invalid}}$ within $R_{\text{rules}}$ $(\alpha = 1, \beta = 0, \gamma = 1, + R_{\text{invalid}})$, the policy no longer collapses to invalid sequences and harmony and diversity return to a reasonable range. However, both metrics are still slightly worse than the full GAPT setting with the complete rule-based term, indicating that the additional rule-based components beyond invalidity checking provide further regularization that improves the tradeoff between coherence and diversity. **Thus, invalidity penalties alone are not sufficient; the full rule-based term further improves the harmony–diversity balance.**
>
> (cont.)

---

> ### Author Response · Authors · 2025-11-21
>
> (continued)
>
> **Overall, these ablations support two conclusions.** **First, GAPT with equal weighting $\alpha = \beta = \gamma = 1$ yields a good balance: it is robust to moderate changes in the coherence weight, while substantially increasing the weight on either the rule-based or adversarial components degrades performance.** **Second, the rule-based invalid output penalty is not only a minor regularizer but plays a critical role in preventing reward hacking via invalid sequences, and the adversarial term must be used in combination with both coherence and rules to achieve the desired Pareto improvement in harmony and diversity.** We will include these findings in the revised paper.
>
> ---
> **Table 7: Ablation study on reward component weighting in Equation 6. We report Harmony (note-in-chord ratio, %) and Diversity (Vendi Score) on the Test Set and Held-out Dataset (Wikifonia). Higher is better for both metrics.**
>
> | System                                                   | Test Set Harmony ↑ | Test Set Diversity ↑ | Held-out Dataset Harmony ↑ | Held-out Dataset Diversity ↑ |
> |----------------------------------------------------------|---------------------|-----------------------|-----------------------------|-------------------------------|
> | GAPT                                                     | 0.497               | 26.645                | 0.470                       | 11.295                        |
> | Upweight Coherence ($\alpha=2, \beta=1, \gamma=1$)       | 0.494               | 26.742                | 0.476                       | 11.553                        |
> | Upweight Rules ($\alpha=1, \beta=2, \gamma=1$)           | 0.475               | 25.667                | 0.458                       | 10.628                        |
> | Upweight Adversarial ($\alpha=1, \beta=1, \gamma=2$)     | 0.456               | 26.317                | 0.449                       | 11.184                        |
> | Exclude Rules ($\alpha=1, \beta=0, \gamma=1$)            | N/A                 | N/A                   | N/A                         | N/A                           |
> | Exclude Rules + Invalid Penalty ($\alpha=1, \beta=0, \gamma=1, +$ $R_{\text{invalid}}$) | 0.488               | 25.072                | 0.461                       | 10.428                        |
>
>
> > What could be the reason for harmonic incoherence (like in Hooktheory Example 4)? Is it because the melody has higher density of notes in beats comparing to other successful accompaniments? When GAPT underperforms harmony (e.g., slight drop on OOD), what musical patterns cause it? Maybe because of the training dataset?
>
> Thank you for this very perceptive question and for carefully listening to the examples.
>
> We believe the incoherence in Hooktheory Example 4 arises mainly from two factors:
>
> **Higher melody density.** The melody in this example has a much higher note density per beat than more typical training and test melodies. Our policy operates in a strictly online, frame based setting without access to future melody tokens. In this regime, dense melodies introduce many local, sometimes conflicting cues (for example passing tones, neighbor tones, and syncopations that span chord boundaries). The policy must respond to these short range patterns before the full phrase is revealed, which increases the chance that the model commits early to an incorrect progression.
>
> **Less common chord progression.** The melody implies a less common progression that is rare in our predominantly pop oriented training corpus. As a result, this progression is supported by only a small subset of training examples, which makes it harder for the policy to correctly anticipate.
>
> Our training data is mostly western popular and folk music, so the chord distribution is dominated by common pop progressions. When the melody implies a rarer progression, the model has less prior exposure to similar trajectories. Furthermore, because the accompaniment policy is strictly online, it does not see future melody tokens and cannot observe the full evolution of the phrase. In these edge cases, it must anticipate the upcoming harmonic intent from limited local context and can commit to a progression that diverges from the intended one.
>
> **We see this example as an illustration of a challenging corner case rather than a typical failure mode.** We are very interested in extending future work to handle such situations better, for instance by improving recognition of repeated patterns like in Example 4, incorporating richer style coverage in the dataset, and developing models that can more robustly predict under ambiguous or high density inputs.
>
> (cont.)

---

> > ### Author Response · Authors · 2025-11-21
> >
> > (continued)
> >
> > > What are end-to-end latency numbers (ms) and GPU budget during live use? How does lookahead/commit horizon trade off with perceived responsiveness?
> >
> > Thank you for raising this practical question, which is very important for real time deployment.
> >
> > In our user study setup, we use an L40S GPU server. Some key statistics of the deployment are:
> >
> > - **Model memory footprint.** The model occupies approximately 2 GB of VRAM on the L40S GPU.
> > - **Backend inference time.** The computation time per request is around 200 to 300 ms, depending on how many frames are committed in a single request (that is, on the requested generation horizon).
> > - **Network latency.** The communication delay depends on network conditions, but in our experiments it remains below 100 ms round trip.
> >
> > Following ReaLJam [1], we adopt a fixed lookahead and commit horizon to trade off robustness against responsiveness. A larger lookahead and commit horizon increases robustness to network jitter and keeps a stable buffer of planned chords in the interface, which reduces audible glitches. However, this comes at the cost of slower perceived reaction to sudden changes in the user’s playing. Guided by the ReaLJam findings and pilot sessions, we choose a commit time of 4 beats and a lookahead time of 4 beats for all user study sessions, which we found to give a smooth and glitch free experience while preserving acceptable responsiveness. **Overall, this configuration provides a practical compromise between stability and responsiveness for real time use.** We will add these numerical latency and compute details to the system description in the revised paper.
> >
> > > Would you mind elaborating a bit more on applying GAPT outside real-time accompaniment?
> >
> > We appreciate this question and the opportunity to clarify the scope of GAPT.
> >
> > In our work, GAPT is instantiated for melody to chord accompaniment, but the underlying idea is more general. Many RL post training pipelines for generative models (for example in LLMs) suffer from similar reward hacking patterns [2], where the policy exploits a learned reward in ways that drift away from the data distribution while still achieving high scalar scores. **We are excited to extend this work to those settings where GAPT can be used as a lightweight, model side regularizer:**
> >
> > - **The discriminator provides an adaptive data prior**, penalizing trajectories that deviate too far from the training distribution even if they satisfy the hand designed or learned reward.
> > - **This adversarial signal can complement or partially replace a KL based constraint**, potentially providing a more data driven and computationally efficient alternative to large scale KL penalties that require tight coupling to a strong reference model.
> >
> > **In summary, while our experiments focus on real time accompaniment, the GAPT framework is designed to be applicable to a broader class of RL post training problems where reward hacking and distribution drift are major concerns.**
> >
> > [1] Scarlatos, A., Wu, Y., Simon, I., Roberts, A., Cooijmans, T., Jaques, N., ... & Huang, A. (2025, April). ReaLJam: Real-Time Human-AI Music Jamming with Reinforcement Learning-Tuned Transformers. In Proceedings of the Extended Abstracts of the CHI Conference on Human Factors in Computing Systems (pp. 1-9).
> >
> > [2] Wan, Y., Wu, J., Abdulhai, M., Shani, L., & Jaques, N. (2025). Enhancing Personalized Multi-Turn Dialogue with Curiosity Reward. arXiv preprint arXiv:2504.03206.

---

> > > ### Comment · Reviewer_pmqm · 2025-11-25
> > >
> > > Grateful to detailed replies, and those comments address my concerns. I increased the rating from 6 to 8.

---

### Author Response · Authors · 2025-11-21
**Additional experiments and analyses added in the revised draft**

We sincerely thank all reviewers for their thoughtful and constructive feedback, which has greatly helped us improve the clarity, experiments, and overall presentation of the paper.

Here is a brief summary of the additional experiments and analyses added in the revised draft:

1. **Ablation on scalar reward weights $\alpha, \beta, \gamma$ and invalid penalty $R_{\text{invalid}}$**, Appendix C.9, Table 7, line 1015. GAPT with equal weights $\alpha = \beta = \gamma = 1$ is robust to moderate coherence upweighting, while strongly upweighting rules or adversarial reward degrades both harmony and diversity, removing rules leads to reward hacking via invalid outputs, and keeping only an invalidity penalty partially stabilizes training but still underperforms the full rule based term.
2. **Three seed stability study for GAPT, GAPT w/o Adv., and ReaLchords**, Appendix C.7, Table 6, line 910. Standard deviations over three RL runs are small for all models and GAPT consistently achieves higher diversity than ReaLchords and the non adversarial ablation at similar or slightly better harmony, indicating low variance and stable gains across random initializations.
3. **Training dynamics and adversarial reward schedule curves**, Figure 8, line 938. The overall scalar reward, adversarial reward, discriminator accuracy, and discriminator loss evolve smoothly over training, and the adaptive update schedule keeps the adversarial reward in a moderate numerical range, supporting the practical stability of the adversarial post training.
4. **Critic value comparison between GAPT and GAPT w/o Adv. and relation to adversarial reward**, Figures 9 and 10, lines 961 and 972. Critic values for GAPT are slightly higher than for GAPT w/o Adv. and the difference closely tracks the additional adversarial reward, showing that the critic successfully estimates the incremental contribution of the adversarial term without destabilizing value learning.
5. **GRPO / Dr.GRPO as an alternative RL objective**, Appendix C.10, Table 8, line 1040. GRPO without adversarial training shows the same diversity collapse as PPO based RL post training, while adding adversarial training on top of GRPO restores diversity and slightly improves harmony on both the test set and the out of distribution dataset, confirming that the benefits of GAPT are robust across different policy gradient algorithms.
6. **Melody conditioned discriminator $D_{\psi}(x,y)$ versus chord only discriminator $D_{\psi}(y)$**, Appendix C.11, Table 9, line 1067. Conditioning the discriminator on both melody and chords still improves diversity over the non adversarial baseline but yields lower diversity and slightly worse held out harmony than the chord only discriminator, suggesting that conditioning on melody makes it easier for the discriminator to overfit specific training pairs and weakens its regularization effect.
7. **Chord vocabulary specification and pitch class statistics to rule out dense chord inflation**, Appendix C.12, text at line 1080 and Table 10, line 1085. We specify a shared vocabulary of 2,821 chord symbols (triads, sevenths, extensions, inversions, and suspensions) and show that the average number of distinct pitch classes per chord for all models (3.00–3.26) is close to the dataset averages (3.28–3.54), indicating that none of the models, including GAPT, artificially inflate the note in chord ratio by predicting unusually dense chords.
8. **Long horizon evaluation at $T = 512$ with naive extension and sliding window inference**, Appendix C.13, Figures 11 and 12, lines 1117 and 1155. For both long sequences from the test set and from Wikifonia the note in chord ratio remains stable over time, and the sliding window strategy with 256 frame context achieves consistently better harmony than naive context extension, showing that training remains stable and GAPT maintains good performance beyond the RL horizon.

---

### Meta-Review · Area_Chair_Y5mQ · 2025-12-23

**Summary:**

The work addresses reward hacking in RL for melody-to-chord accompaniment by incorporating adversarial learning to strengthen the reward signal, particularly under incomplete human-defined rewards and out-of-distribution samples. The key idea is to augment the RL reward with a discriminator-based term. Experiments show strong performance in both quantitative metrics and perceptual evaluations. All reviewers appear satisfied with the rebuttal, and there are no major outstanding concerns. Therefore, I recommend acceptance.

**Reviewer Concerns:**

- Reviewer pmqm asked for ablation of weightings for the proposed overall reward.

- Reviewer SxDd’s main concern is training stability due to adversarial discriminator learning.

-  Reviewer VfzC raised concerns about the lack of comparison to RL post-training approaches.
- Reviewer wu95 mainly has two concerns (i) experiments are confined to melody–chord accompaniment and short sequences (T≤256 sixteenth-note frames), leaving scalability to longer timesteps and stability at larger T untested; (ii) Metrics are simplistic (note-in-chord ratio) and reported gains over baselines are small or mixed (e.g., Online MLE matches/exceeds GAPT on most Table 2 metrics), so the quantitative results don’t convincingly establish efficacy.

**Reviewer Scores:**

- The concern on weights ablation is addressed by additional experiments: Ablations show that equal-weight GAPT is the most robust/balanced setting, and that rule-based invalid penalties are essential to prevent reward hacking, while the adversarial term only helps when combined with both coherence and rules to jointly improve harmony and diversity. Also, the reviewer was satisfied with the authors’ responses and increased their score from 6 to 8.

- The training instability concern was addressed with additional experiments: techniques such as discriminator warm-up or update scheduler appear to improve stability, and the evidence suggests training is stable in practice. The reviewer also seemed satisfied with the response and raised their score.

- The authors addressed this by adding a GRPO baseline and demonstrating overall superior performance across metrics, including harmony quality and diversity. The reviewer also seemed satisfied with the response and raised their score.

- The authors added additional experiments (Figs. 11–12) extending the RL sequence length to (T=512), showing that training remains stable and that GAPT maintains strong performance even on sequences longer than the original RL horizon. Although the reviewer did not respond before the rebuttal window closed, I believe the authors have addressed most of the reviewer’s concerns.

---

### Decision · Program_Chairs · 2026-01-26

Accept (Poster)